# Quantification of carbon monoxide emissions from African cities using TROPOMI

**Gijs Leguijt[1,2], Joannes D. Maasakkers[1], Hugo A. C. Denier van der Gon[2], Arjo J. Segers[2], Tobias Borsdorff[1], and Ilse Aben[1,3]**

[1]SRON Netherlands Institute for Space Research, Leiden, the Netherlands
[2]Department of Climate, Air and Sustainability, Netherlands Organisation for Applied Scientific Research, TNO, Utrecht, the Netherlands
[3]Department of Earth Sciences, Vrije Universiteit Amsterdam, Amsterdam, the Netherlands

**Correspondence:** Gijs Leguijt (g.leguijt@sron.nl)

**Abstract.** Carbon monoxide (CO) is an air pollutant that plays an important role in atmospheric chemistry and is mostly emitted by forest fires and incomplete combustion in, for example, road transport, residential heating, and industry. As CO is co-emitted with fossil fuel $CO_2$ combustion emissions, it can be used as a proxy for $CO_2$. Following the Paris Agreement, there is a need for independent verification of reported activity-based bottom-up $CO_2$ emissions through atmospheric measurements. CO can be observed daily at a global scale with the TROPO-spheric Monitoring Instrument (TROPOMI) satellite instrument with daily global coverage at a resolution down to $5.5 \times 7\,km^2$. To take advantage of this unique TROPOMI dataset, we develop a cross-sectional flux-based emission quantification method that can be applied to quantify emissions from a large number of cities, without relying on computationally expensive inversions. We focus on Africa as a region with quickly growing cities and large uncertainties in current emission estimates. We use a full year of high-resolution Weather Research and Forecasting (WRF) simulations over three cities to evaluate and optimize the performance of our cross-sectional flux emission quantification method and show its reliability down to emission rates of $0.1\,Tg\,CO\,yr^{-1}$. Comparison of the TROPOMI-based emission estimates to the Dynamics–Aerosol–Chemistry–Cloud Interactions in West Africa (DACCIWA) and Emissions Database for Global Atmospheric Research (EDGAR) bottom-up inventories shows that CO emission rates in northern Africa are underestimated in EDGAR, suggesting overestimated combustion efficiencies. We see the opposite when comparing TROPOMI to the DACCIWA inventory in South Africa and Côte d'Ivoire, where CO emission factors appear to be overestimated. Over Lagos and Kano (Nigeria) we find that potential errors in the spatial disaggregation of national emissions cause errors in DACCIWA and EDGAR respectively. Finally, we show that our computationally efficient quantification method combined with the daily TROPOMI observations can identify a weekend effect in the road-transport-dominated CO emissions from Cairo and Algiers.

## 1 Introduction

Carbon monoxide (CO) is an air pollutant that is mostly emitted by anthropogenic sources. It is a product of incomplete combustion in, for example, road transport, residential heating, industry, and forest fires (Zhong et al., 2017). CO is a precursor of ozone, and because it reacts with the hy-droxyl radical (OH), its presence effectively increases the atmospheric lifetime of methane (Daniel and Solomon, 1998; Jacob, 1999; Wuebbles and Hayhoe, 2002). The concentration of CO is therefore important for climate modeling. Furthermore, as many processes that emit CO also emit carbon dioxide ($CO_2$), knowledge of CO emission rates can provide additional information about $CO_2$ emissions (Wu et al., 2022;

Park et al., 2021). The Intergovernmental Panel on Climate Change (IPCC) identified a need for independent verification of the reported greenhouse gas emissions through measurements (IPCC, 2019). From space this is challenging for $CO_2$, as its long atmospheric residence time results in high background concentrations, making it hard to detect emissions. For this reason, measuring the "short lived" CO can be a useful alternative (Silva et al., 2013; MacDonald et al., 2023). We present a method to quantify CO emission rates over cities in Africa using TROPOspheric Monitoring Instrument (TROPOMI) satellite observations.

As transport (23 %) and residential heating (35 %) are key contributors to total anthropogenic CO emissions (Zhong et al., 2017), cities are an important source of CO. In Africa, the contributions of transport and residential heating are estimated at 27 % and 38 % of anthropogenic CO emissions respectively in the DICE-Africa inventory (Marais and Wiedinmyer, 2016) and 17 % and 72 % in the Dynamics–Aerosol–Chemistry–Cloud Interactions in West Africa (DACCIWA) inventory (Keita et al., 2021). The importance of these two sectors is further confirmed by a large number of ground-based measurements specifically aimed at traffic (Diab et al., 2005; Lindén et al., 2008; Zakari et al., 2020; Doumbia et al., 2021) and domestic heating (Havens et al., 2018; Kansiime et al., 2022; Saleh et al., 2023) that show CO concentrations in African cities exceeding air quality guidelines by the World Health Organization. Urbanization scenarios predict a growth in both the number of megacities and their populations, leading to larger emission rates and increased health risks. Africa is predicted to have a large urbanization rate in the coming years. Hoornweg and Pope (2017) predict the continent to house 5 of the 10 largest cities by 2100, compared to 1 of 10 today. Africa is also a region for which relatively large uncertainties are present in emission inventories, as only a few are dedicated to the region (Keita et al., 2021). Current emission inventories are based on so-called bottom-up methods, where emissions are estimated by combining activity data (e.g., national fuel consumption statistics) with emission factors and spatially distributing the emission estimates using proxies like population density (Janssens-Maenhout et al., 2019). These bottom-up methods are also used to report country-level greenhouse emission estimates to the United Nations Framework Convention on Climate Change (UNFCCC). However, lack of detailed data results in large uncertainties (Macknick, 2011; Cai et al., 2019; Oda et al., 2019).

Independently of bottom-up methods, emissions can also be estimated by top-down methods, where atmospheric concentrations are measured and used to infer the corresponding emission rates. Multiple studies have investigated urban CO emissions using ground-based measurements (Badarinath et al., 2007; McKain et al., 2012; Bi et al., 2022). Many studies have also shown the capability of satellite measurements for this specific task (Borsdorff et al., 2020; Tian et al., 2022a;

Plant et al., 2022; Wu et al., 2022). For CO, the TROPOspheric Monitoring Instrument (TROPOMI) on ESA's Sentinel 5 precursor satellite is of particular interest (Veefkind et al., 2012). It was launched in 2017 and provides daily global coverage with a resolution of $5.5 \times 7\,\text{km}^2$, which makes it suited to investigate city emissions worldwide.

An advantage of polar-orbiting satellites is their ability to monitor the entire globe. However, most satellite-based studies of CO so far have focused either on regional inversions (Yumimoto et al., 2014; Qu et al., 2022) or on trends in concentrations (Lama et al., 2020; Park et al., 2021; Hedelius et al., 2021), while only a few studies have tried to quantify emissions from individual cities or point sources (Dekker et al., 2017; Borsdorff et al., 2020). These urban emission quantifications use atmospheric inversions, which require computationally expensive high-resolution simulations with chemical transport models (CTMs). Although inversions are able to get relatively accurate emission estimates, they are difficult to apply to a large number of sources. To take full advantage of the TROPOMI data, we adjust the mass balance cross-sectional flux (CSF) method, originally developed for high-resolution point-source quantifications, to be used with TROPOMI data over urban areas. After evaluating the method using atmospheric transport simulations, we use it to estimate emissions from the largest cities in Africa.

## 2   Data and methods

This section describes the different data products used in the development of the cross-sectional flux method and the simulations that were used to calibrate the model and evaluate its performance using an observing system simulation experiment (OSSE). An OSSE is an experiment where a model or method is applied to synthetic data to evaluate the benefit of using this data and/or method. Which for this work means evaluating whether the CSF can be used to correctly estimate emissions from TROPOMI-like synthetic data. Figure 1 shows the roles of the different data products that are used and further described in Sect. 2.1 to 2.6. In addition, in Sect. 2.6 we show that the CSF method can be successfully applied to simulated data.

### 2.1   TROPOMI carbon monoxide data product

TROPOMI provides total column carbon monoxide concentrations with daily global coverage at 13:30 local time using the shortwave-infrared band (SWIR) at 2305–2385 nm (Veefkind et al., 2012). From the spectral signal, the CO concentration is inferred using the shortwave-infrared CO retrieval (SICOR) algorithm (Borsdorff et al., 2018). We use 3 years of data (2019–2021) from the operational data product (Landgraf et al., 2018). To assure high quality data, all pixels with a TROPOMI quality flag below 0.7 are removed, leaving data that are cloud-free or only have low altitude clouds. The CO concentration over cloud-free water surfaces

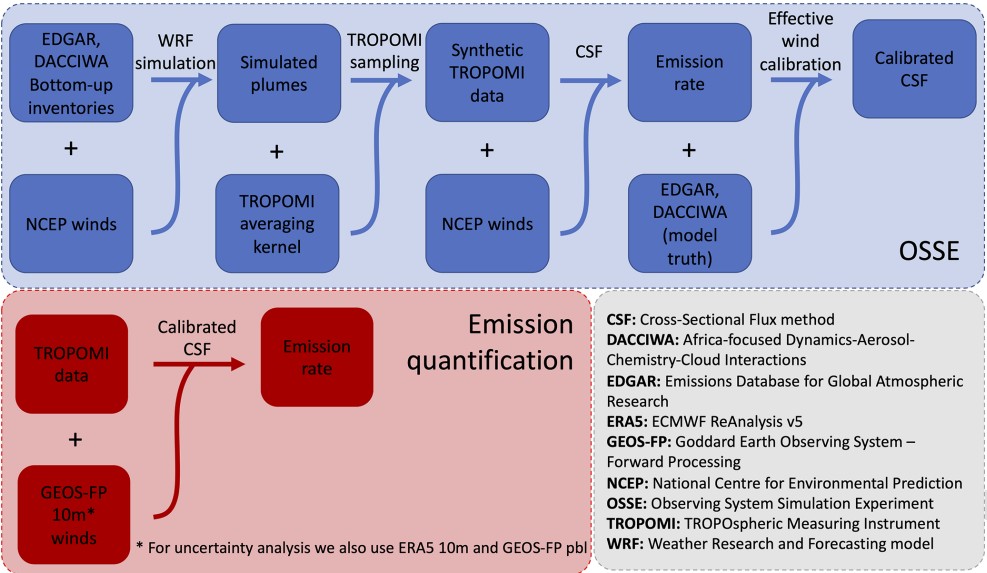

**Figure 1.** Schematic description of the use of the different data products within the OSSE and the subsequent emission quantification using TROPOMI data. The data products are discussed in Sect. 2.1–2.6. First, the CSF is applied to simulated synthetic plumes in order to determine appropriate values for the various parameters used in our method. Second, an effective wind is calibrated by using the known emission rates of the simulated plumes following the procedure by Varon et al. (2018). The CSF, now calibrated on the synthetic plumes, is subsequently applied to satellite data to estimate emission rates of African cities.

is difficult to retrieve due to the low intensity of reflected light; therefore, we only use observations with a quality flag equal to 0.7 (low altitude clouds) over water. The resulting dataset shows good agreement with ground-based measurements, with a mean difference per station of $2.45 \pm 3.38\%$ to the unscaled Total Carbon Column Observing Network (TC-CON, Wunch et al., 2011) columns and $6.5 \pm 3.54\%$ to the Infrared Working Group of the Network for the Detection of Atmospheric Composition Change (NDACC-IRWG, 2023) measurement stations (Sha et al., 2021).

## 2.2 EDGAR and DACCIWA bottom-up inventories

We use two different bottom-up inventories to compare the TROPOMI emission estimates: the Emissions Database for Global Atmospheric Research (EDGAR) version 5 (Oreggioni et al., 2021) and the Africa-focused Dynamics–Aerosol–Chemistry–Cloud Interactions in West Africa (DACCIWA) inventory (Keita et al., 2021). These inventories are also used in our atmospheric transport simulations (Sect. 2.3) to simulate TROPOMI observations. Both inventories provide yearly gridded emission rates at 0.1° resolution up to 2015. Due to its global scope, the EDGAR inventory relies mostly on international statistics and spatial proxies combined with national data, while its emission factors are based on IPCC methodology for greenhouse gases (Eggleston et al., 2006) and the EMEP/EEA emission inventory guidebook for air pollutants (Nielsen, 2013). The DACCIWA inventory provides emission rates over the

African continent, ranging from $-35$ to $38°$ latitude and $-25.5$ to $63.5°$ longitude. It uses similar international data but is supplemented by local measurements of emission factors and data from local authorities (Keita et al., 2021). As the DACCIWA inventory characterizes emission from fewer (sub)sectors than EDGAR, we merge different sectors in EDGAR to match those used in the DACCIWA inventory to make them intercomparable. When reporting urban emissions from EDGAR and DACCIWA, we sum emissions over the pixel closest to the city center, its eight neighbors, and all directly attached pixels where the population density exceeds the surroundings by 1.8 standard deviation. Changing the city mask to $0.3 \times 0.3°$ or $0.7 \times 0.7°$ boxes changes the emissions by 10 %–20 % for 17 out of 29 cities in EDGAR and 16 out of 29 cities in DACCIWA. Although larger deviations up to 50 % in densely populated areas like South Africa are observed, the observed patterns discussed in Sect. 3 are valid for these masks as well.

## 2.3 WRF chemical transport model

To test and calibrate our emission quantification approach, we apply our CSF method to simulated TROPOMI data for three urban areas. We use the Weather Research and Forecasting (WRF) chemical transport model version 4.1 (Powers et al., 2017) to simulate column CO mixing ratios over Cairo (Egypt), Bamako (Mali), and Lagos (Nigeria) for 2019, using December 2018 as the spin-up month. These three African cities form a diverse set, with Cairo next to the Nile river,

Bamako at the boundary of the Sahara desert, and Lagos at the coast of the Atlantic Ocean. We simulate CO as an inert tracer and drive the simulations with meteorological fields from the National Center for Environmental Prediction (NCEP, 2000). All simulations have three-layer nested domains, where the outer domain covers $2673 \times 2673 \, \mathrm{km}^2$ at a resolution of 27 km; the middle and inner domains cover $891 \times 891 \, \mathrm{km}^2$ at 9 km resolution and $315 \times 315 \, \mathrm{km}^2$ at 3 km resolution respectively (Fig. 2). Initial and 6-hourly boundary conditions to capture the background CO are taken from the Copernicus Atmosphere Monitoring Service (CAMS) at $0.25° \times 0.25°$ resolution (Inness et al., 2015). The resulting background is scaled to match the mean background observed by TROPOMI over the full year. We use emissions from the global Emissions Database for Global Atmospheric Research (EDGAR) version 5 and the Africa-focused Dynamics–Aerosol–Chemistry–Cloud Interactions in West Africa (DACCIWA) inventory distributed across the vertical model levels according to the sector-specific vertical profiles provided by Bieser et al. (2011). Typical injection heights for CO emissions from transport and the residential sector are 0–20 m, while emissions from industry are typically injected into the atmosphere at 100–200 m (Bieser et al., 2011). City-specific hourly, daily, and monthly temporal profiles for each emission sector are taken from Guevara et al. (2021). To maintain flexibility over model output, the different sectors in the emission inventories (19 for EDGAR and 6 for DACCIWA) are simulated separately. We sample the model output (at the TROPOMI overpass time) to facilitate comparison to the TROPOMI carbon monoxide data as discussed in detail in Sect. 2.6.

While EDGAR and DACCIWA only include the primary production of CO, the concentrations observed by TROPOMI include CO from secondary production as well. CO is produced by oxidation of volatile organic compounds (VOCs), with methane as the main contributor (Rozante et al., 2017). Mixing ratios of non-methane volatile organic compounds (NMVOCs) observed in urban locations are typically of the order of $10^{-3}$–$10^{-2}$ [NMVOC]/[CO] (Von Schneidemesser et al., 2010). Dekker et al. (2019) showed that chemical production of CO by methane and NMVOC over cities only contributes 4 % to the total CO signal, justifying the simulation of CO as an inert tracer in our approach. Due to the 10-year atmospheric lifetime of methane, its contribution to CO production will result in a uniform concentration (Park et al., 2013) that is subtracted with the background. NMVOCs have lifetimes of 0.6–10 d (Guo et al., 2007) that are much shorter than the lifetime of $CH_4$, but due to their low urban mixing ratios ($\sim 1 \%$), their effect on the estimated emission rate is much smaller than the reported uncertainty of the CSF. This is consistent with the observation that the emission estimates of individual transects (that span a timescale of up to $\sim 10 \, \mathrm{h}$) are stable and do not increase with increasing distance from the city (Sect. 2.4).

## 2.4 Cross-sectional flux method

The cross-sectional flux method (CSF) has been shown to be an effective way to quantify emission rates of plumes observed by satellites (Varon et al., 2018, 2020; Sadavarte et al., 2021b; Tian et al., 2022b). It is based on the continuity equation, which relates the flux through a closed surface to the associated emission rate:

$$Q = \oint U_\perp \Delta\Omega \mathrm{d}A, \tag{1}$$

where $Q$ ($\mathrm{kg\,s^{-1}}$) is the emission rate, $U_\perp$ ($\mathrm{m\,s^{-1}}$) is the wind speed perpendicular to the closed surface, $\Delta\Omega$ ($\mathrm{kg\,m^{-3}}$) is the enhancement at the closed surface, and $\mathrm{d}A$ ($\mathrm{m}^2$) is a surface element. As illustrated in Fig. 3a, the plumes have a distinct direction as they move with the wind, they are very directional, and it suffices to integrate over perpendicular transects that cover the entire plume width (Fig. 3b). Equation (1) can then be rewritten as

$$Q = \int U_\perp(x, y)\Delta\Omega(x, y)\mathrm{d}y, \tag{2}$$

with $x$, $y$ coordinates along and perpendicular to the plume respectively as in Varon et al. (2018). Assuming a constant emission rate, transects at different distances downwind of the source should yield the same emission quantification and can be averaged to make the method more robust.

We then optimize the implementation of the CSF on TROPOMI data for city-like sources. Figure 3a shows a CO plume observed by TROPOMI over Cairo on 7 April 2019. As expected, the plume follows the 10 m wind direction given by NASA/GMAO GEOS-FP reanalysis data (Molod et al., 2012). We start by determining the city center for our purposes defined as the location at the center of the urban emissions and therefore best representing the origin of the city's total emission. The location of the center is determined by taking the weighted average position of the pixels in DACCIWA that are part of the city mask introduced in Sect. 2.2. Weights of the pixels are equal to their emission rate. By determining the city location using the emission inventory, we ensure that we are comparing similar regions when we compare our satellite-based emission estimates with the emission inventories in Sect. 3. To make sure the entire plume is downwind, we start the transects of our CSF 0.1° upwind of this city center. As the wind direction is an important source of uncertainty, the downwind direction can not be solely based on the GEOS-FP reanalysis wind data. Instead, following Sadavarte et al. (2021a), we infer the wind direction from the satellite observations by selecting the direction of the highest mean downwind concentration within 90° of the reanalysis wind direction. To do so, we calculate the mean downwind concentration over 180 boxes (0.1° width and 0.4° length) rotated at 1° intervals and pick the direction with the highest downwind concentration. In the absence of a clear plume, this method would create a positive bias, as it would select

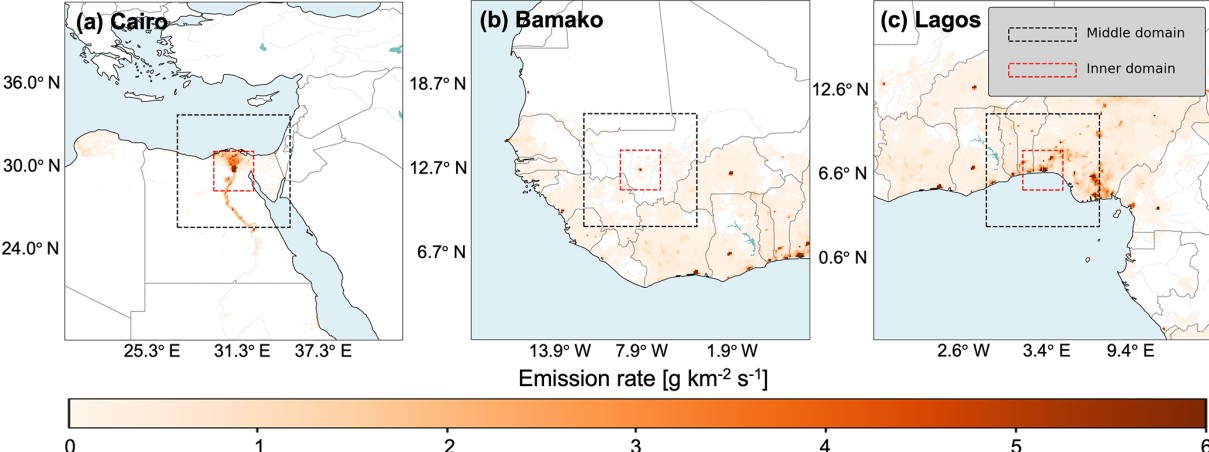

**Figure 2.** Domain setup of the WRF simulations over Cairo (Egypt), Bamako (Mali), and Lagos (Nigeria). The inner domain (red) spans $315 \times 315$ km$^2$ around the city at 3 km resolution. The middle (black) and outer (full figure) domain cover $893 \times 893$ and $2673 \times 2673$ km$^2$ at 9 and 27 km resolution respectively. All panels show the emission rates from the DACCIWA inventory.

the highest enhancement in the noise. Therefore, we use the reanalysis wind direction if the mean enhancement does not exceed 5 ppb. To calculate CO enhancements, we subtract a background calculated as a mean upwind concentration over a $0.4° \times 0.4°$ square starting $0.3°$ upwind from the city center (Fig. 3b). If this box contains fewer than five valid pixels, we extend it symmetrically with two arcs of a circle of 10, 20, 45, up to $60°$ until there are at least five TROPOMI pixels in the background region (Fig. 3c). We use extension in an arc-like fashion rather than increasing the size of the square to be able to get estimate background values for coastal cities. Retrievals over water are only possible if there are clouds present; hence, increasing the size of the background square upwind could result in background pixels far away from the city that are not representative for the local background.

After determining the initial direction of the plume, we need to better capture the shape of the plume to draw transects perpendicular to the plume. The shape of the plume is determined in two steps. First, we select all pixels in a downwind box ($0.3°$ width, $0.8°$ length) that exceed the mean concentration in the surrounding $3° \times 3°$ area by more than 1.8 standard deviations; these pixels are referred to as the spline mask. We then fit a 2D-spline ($0.8°$ length) through the resulting spline mask. If, due to a lack of signal or missing pixels, the spline mask contains fewer than three pixels, a spline fit is unlikely to capture the true plume shape, and we use a straight line in the (optimized) wind direction instead (Fig. 3c). The transects ($0.4°$ width) are drawn perpendicular to the spline, separated by $0.04°$. The transects have a larger width than the box used to determine the spline mask to ensure the transects cover the entire plume width. All pixels overlapping with the transects are used in the emission quantification. We also include days where no clear plume is visible to avoid systematic overestimation of the average emission rate. Using Eq. (2), an emission estimate can be de-

rived for every transect. We stop drawing transects when the emission rate estimates of two consecutive transects are more than 1 standard deviation below the mean estimate of the earlier transects, indicating the end of the plume. Transects with less than 70 % pixel coverage are removed from the estimate, as they will not have a complete integral, resulting in underestimated emissions.

Contrary to studies using high-resolution satellites (Varon et al., 2018, 2020), the plumes observed with TROPOMI cover distances over which there can be significant fluctuations in wind speed and direction. We therefore use the wind speed at each transect instead of a single wind speed for the entire plume. The wind speed at the transects is calculated in two steps. First, a wind speed is calculated for each TROPOMI pixel by interpolation of the reanalysis wind product. Second, the wind speed for each transect is determined by taking the average wind speed of the overlapping TROPOMI pixels, weighted by the length of the overlap. Similar to trends observed in Sadavarte et al. (2021b), the first two transects are found to have roughly 30 % lower emissions than the transects further away, which have a stable mean emission rate. This pattern is consistent across the cities investigated. One reason is that the early plume only captures part of the city's emissions, another explanation is that the associated pixels might see a partial-pixel absorption saturation effect (Pandey et al., 2019). Incorporating the first two transects would result in an average underestimation of emissions by 8 %. We therefore remove the first two transects from the emission estimation.

On the spatial scale relevant to plumes observed by TROPOMI, there can be contamination of the city signal by carbon monoxide produced by open fires (e.g., agricultural fires or wildfires). CO enhancements caused by open fires can result in overestimation of either the background or the downwind urban enhancement, depending on their location.

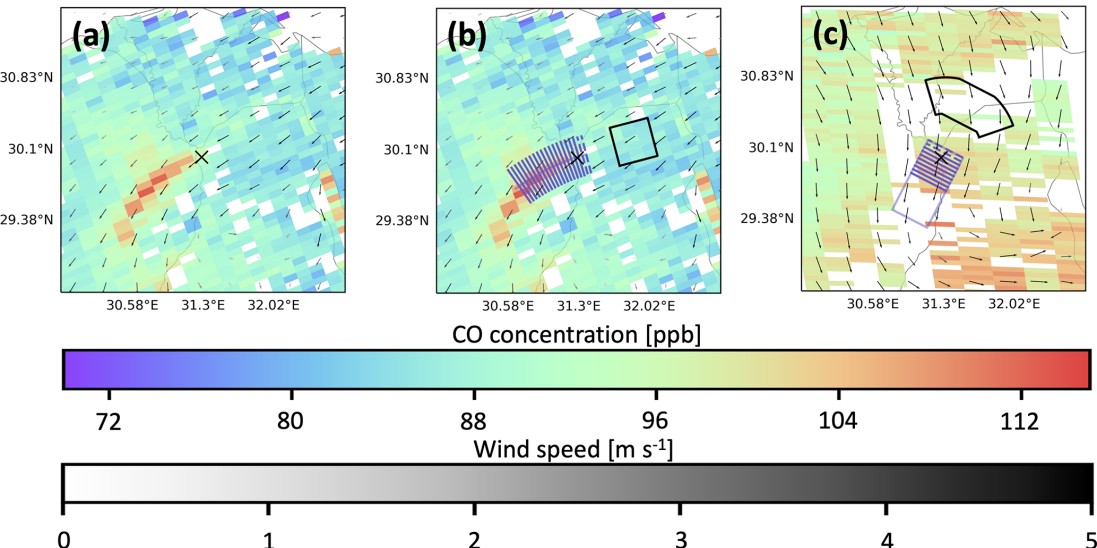

**Figure 3.** Example of how the cross-sectional flux transects perpendicular to the plume are drawn. **(a)** TROPOMI data over Cairo on 7 April 2019. The city center is shown with a cross and taken from the DACCIWA emission inventory. The arrows show direction and magnitude of GEOS-FP 10 m winds at their native $0.25° \times 0.3125°$ resolution (Molod et al., 2012). **(b)** Pixels downwind of the city that surpass the regional background by more than $1.8\sigma$ form a plume mask through which a 2D spline is fitted (grey line). The transects used for quantification (purple lines) are drawn perpendicular to the spline fit starting $0.1°$ upwind of the city center. The first two transects are dashed to reflect that they are not used for the emission quantification. The background is estimated over the black $0.4° \times 0.4°$ box upwind. **(c)** TROPOMI observation over Cairo on 27 March 2020. Due to a lack of coverage there are insufficient pixels to generate a reliable spline mask. A rectangular box (grey) is therefore used to draw transects instead. The basic background is extended symmetrically, with circle arcs to compensate for a lack of coverage upwind.

To avoid this, days with considerable CO contributions from open fires have been removed from our estimates. These days were selected based on the fire emission data from the Global Fire Assimilation System fire emission (GFAS) inventory (Kaiser et al., 2012) that is based on satellite measurements of fire radiative power. Days with cumulative fire emissions over $57\,\mathrm{Mg\,h^{-1}}$ (equivalent to $0.5\,\mathrm{Tg\,yr^{-1}}$) within $1.5°$ from the city center are removed. Additionally, days with strong burning events closer to the city ($23\,\mathrm{Mg\,h^{-1}}$ within a $0.75°$ radius) are removed as well (Appendix B). Although the change in emission rate by this filtering is limited for most cities, the filter can change estimated emission rates by up to 47 %, as seen in Lusaka (Zambia).

## 2.5 Uncertainty analysis

To estimate the uncertainty of the estimated emission rates, we compile an ensemble of emission estimates for each city. We generate the ensemble by varying parameters of the quantification method such as the wind database used. For example, we vary parameters such as the number of transects and the distance of the background box. To incorporate the uncertainty on the wind data, we use our method with GEOS-FP 10 m altitude winds, GEOS-FP planetary boundary layer (PBL) averaged winds (Molod et al., 2012) as well as 10 m altitude winds from the ERA5 product, provided by the European Centre for Medium-Range Weather Forecasts

(ECMWF; Hersbach et al., 2020). A complete list of the varied parameters and their ranges can be found in Appendix A. For each city, the spread in the resulting ensemble is reported as uncertainty.

## 2.6 Calibration and validation

This section describes the application of the CSF to simulated CO column mixing ratios. The simulations are used to determine parameter settings (e.g., spline length and transect width) and to calibrate an effective wind (Varon et al., 2018) for TROPOMI-sized pixels. In addition, the simulations are used to evaluate how well the CSF can quantify emission rates of simulated plumes.

As simulated (and TROPOMI observed) plumes stay within the inner domain, only the inner domain is used to test the performance of the CSF. A set of synthetic TROPOMI observations is created by sampling the simulation output over the TROPOMI footprints, applying its averaging kernel, selecting pixels based on quality value as discussed in Sect. 2.1, and adding Gaussian noise with a standard deviation equal to the reported uncertainty of the respective TROPOMI pixel. The TROPOMI quality value filtering ensures relatively clear sky observations with good surface sensitivity. We also calculate "idealized" pressure weighted columns, which assume a uniform vertical sensitivity (flat averaging kernel), over the TROPOMI footprints without tak-

ing into account whether there is a valid TROPOMI observation as a first check to see whether the CSF can reproduce the emissions used as model input.

We first test the validity of the CSF method using the idealized columns with 10 m winds output by the WRF simulation. The WRF winds are directly responsible for transport within the simulation and can therefore be considered as the true wind fields behind the modeled concentrations. Parameters like the number of transects and distance of the background region are tuned to get optimal quantification estimates on the simulated data, such that the fitted splines capture the observed curvature of the plumes and the background is not affected by the urban emissions. A list of the different parameters and their values can be found in Appendix A. While the true wind field varies with altitude, the CSF method requires just a single (2D) wind field that is representative for the transport of the plume. We use the simulations to calibrate the CSF by introducing an effective wind speed that replaces the wind speed in Eq. (2), following the procedure by Varon et al. (2018). The effective wind speed is the wind that best captures the transport of the plume. It is a parametrization of the true wind speed to account for the effects of turbulence and variation in vertical wind speed and injection height. As the emission rates in the WRF simulations are known, the effective wind can be calculated explicitly for every orbit for each of the simulated cities. Figure 4 shows the relation between the effective wind ($U_{eff}$) and the WRF 10 m winds $U_{10}$ averaged over the plume; the fitted linear relation is

$$U_{eff} = a_{10}U_{10} + b_{10}, \qquad (3)$$

with $a_{10} = 1.43$ and $b_{10} = -0.92 \, \mathrm{m \, s^{-1}}$ ($R^2 = 0.82$). $U_{10}$ is the wind speed at the time of overpass at 10 m altitude. We determine the effective wind relationship separately for the planetary boundary layer averaged winds, which tend to be higher than the surface winds. The resulting calibration gives $a_{PBL} = 0.98$ and $b_{PBL} = -0.20$ ($R^2 = 0.62$). While the absolute value of the PBL winds is closer to the effective wind speeds, using the $U_{10}$ winds captures more of the variability.

After determination of the effective wind on plumes with idealized pressure profiles, we test the performance of the CSF on more realistically sampled plumes, which include the TROPOMI quality filtering and averaging kernel sensitivities as described in Sect. 2.1, to see whether the CSF can correctly quantify emission rates from synthetic observations with quality filtering and non-uniform vertical sensitivity. To test the method's sensitivity, we perform an additional effective wind calibration on these data. The resulting linear fit ($a = 1.42$, $b = -0.86$, $R^2 = 0.27$) yields similar results and shows that the filtering has limited impact on the calibration, while the lower $R^2$ value reflects the larger variation in estimated emission rates. At the same time, we test the lower limit to which we can trust the resulting emission estimates, as smaller enhancements are more difficult to distinguish from the background. As the modeled output concentration from the WRF simulations without chemistry scales linearly with the magnitude of the input emissions, emissions from the different sectors provided by the bottom-up inventories can be scaled up and down without having to rerun the chemical transport model. This allows us to simulate plumes from cities with different emission rates with limited effect on the simulated background through scaling of the emission sector most concentrated in the considered urban area. We use this to determine the lower limit to which our method can be trusted. Figure 5 shows a comparison between the simulated input and the retrieved emission rates for the three simulated cities. The results suggest that the CSF is able to reproduce input of the WRF simulations when using 1 year of data for cities with emission rates larger than 0.1 Tg yr$^{-1}$.

To quantify TROPOMI plumes over all major cities in Africa, we will use the NASA/GMAO GEOS-FP wind fields (Molod et al., 2012) rather than the WRF-simulated wind fields, which are available only for the three cities selected for evaluation and calibration. For each TROPOMI pixel, we spatially interpolate the GEOS-FP wind field, which has a $0.25° \times 0.3125°$ spatial resolution and a 1 h time resolution. The NCEP winds that drive the WRF-simulated wind fields have a coarser time resolution of 6 h. Figure 6 shows emission estimates of simulated data using the GEOS-FP wind fields instead of the WRF fields to mimic uncertainties in the wind fields. Individual days are shown as colored dots, while the mean over the full year is shown as a colored line to represent the average estimate. The uncertainty of the average is determined as discussed in Sect. 2.5 and shown as a shaded area. The true emission is shown as a dotted black line and lies within the uncertainty of the estimate for both Cairo and Bamako. For Lagos, the emission rate is underestimated when using the WRF simulations, as the NCEP wind fields that drive the simulations are higher than both the GEOS-FP and ERA5 wind products specifically over Lagos by about 60 %. The difference between the wind products might be caused by the fact that Lagos lies in the West African monsoon region, where transport has been shown to be difficult to model (Liu et al., 2014).

## 3 Results and discussion

After verification of the validity and calibration of the method, we apply it to 29 of the largest cities in Africa. These cities are chosen based on their population or because they are emitting above the CSF's quantification threshold in the DACCIWA inventory. Figure 7 shows the results of our TROPOMI quantification and a comparison with the DACCIWA and EDGAR inventories. These data are also included in Appendix D in Table D1. As discussed in Sect. 2.2, we used different sizes for the city masks applied to the bottom-up inventories to ensure a fair comparison to the satellite-based emission estimates and found the choice of city mask did not impact our conclusions.

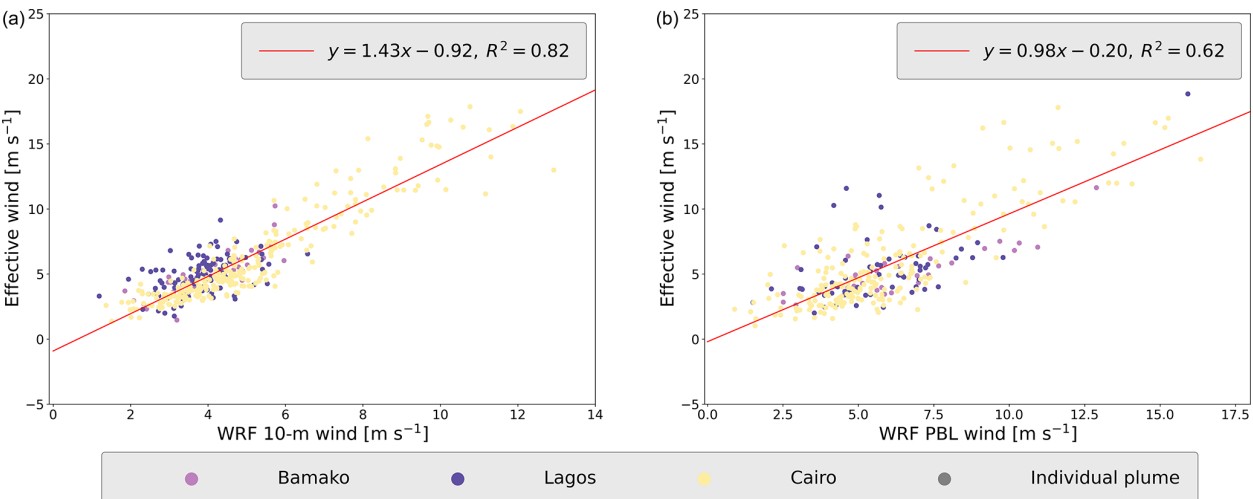

**Figure 4.** Determination of the relation between the effective wind and both the wind speed at 10 m altitude **(a)** and the planetary boundary layer averaged winds **(b)**. The effective wind corrects for the effects of turbulence, injection height, and variation in the vertical wind profile. The simulated plumes used in the calibration cover a full year over Cairo, Bamako, and Lagos.

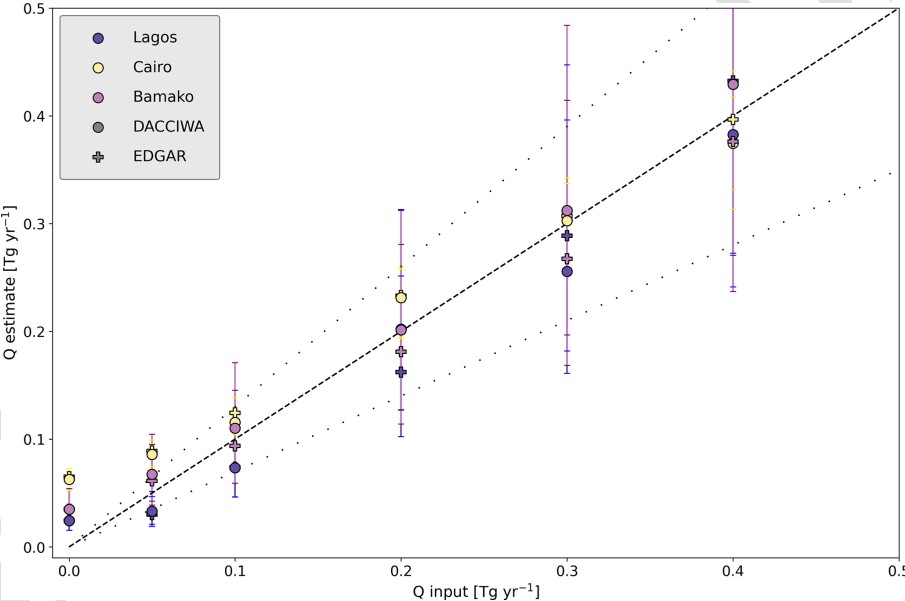

**Figure 5.** Emission quantification by the CSF on simulated plumes. The plumes are simulated with the WRF model for the year 2019 over Lagos (Nigeria), Cairo (Egypt), and Bamako (Mali). The simulations either used the EDGAR global bottom-up inventory or the DACCIWA inventory. The dotted lines show a 30 % deviation from the (dashed) 1 : 1 line.

On average we find TROPOMI emissions of $0.25\,\mathrm{Tg\,yr^{-1}}$ per city, compared to $0.35\,\mathrm{Tg\,yr^{-1}}$ in DACCIWA and $0.18\,\mathrm{Tg\,yr^{-1}}$ in EDGAR. Except for Abuja (Nigeria) and Khartoum (Sudan), the DACCIWA emission estimates are consistently higher than the EDGAR estimates. Additionally, the two inventories disagree on the sectoral breakdown of the emission estimates, with the domestic sector contributing 59 % to the total emission rate in DACCIWA, while EDGAR attributes 54 % of total emissions to the industry sector. For 10 cities TROPOMI and DACCIWA agree within the TROPOMI uncertainty, that is the case for 9 cities in EDGAR. For 16 cities, the TROPOMI estimates are closer to DACCIWA than to EDGAR. The largest differences between TROPOMI and DACCIWA are found for Abidjan (627 %) and Lagos (417 %), while estimates for Cairo (2 %) and Antananarivo (9 %) agree best. To explain the differences between TROPOMI and the inventories, we will now focus on some specific areas.

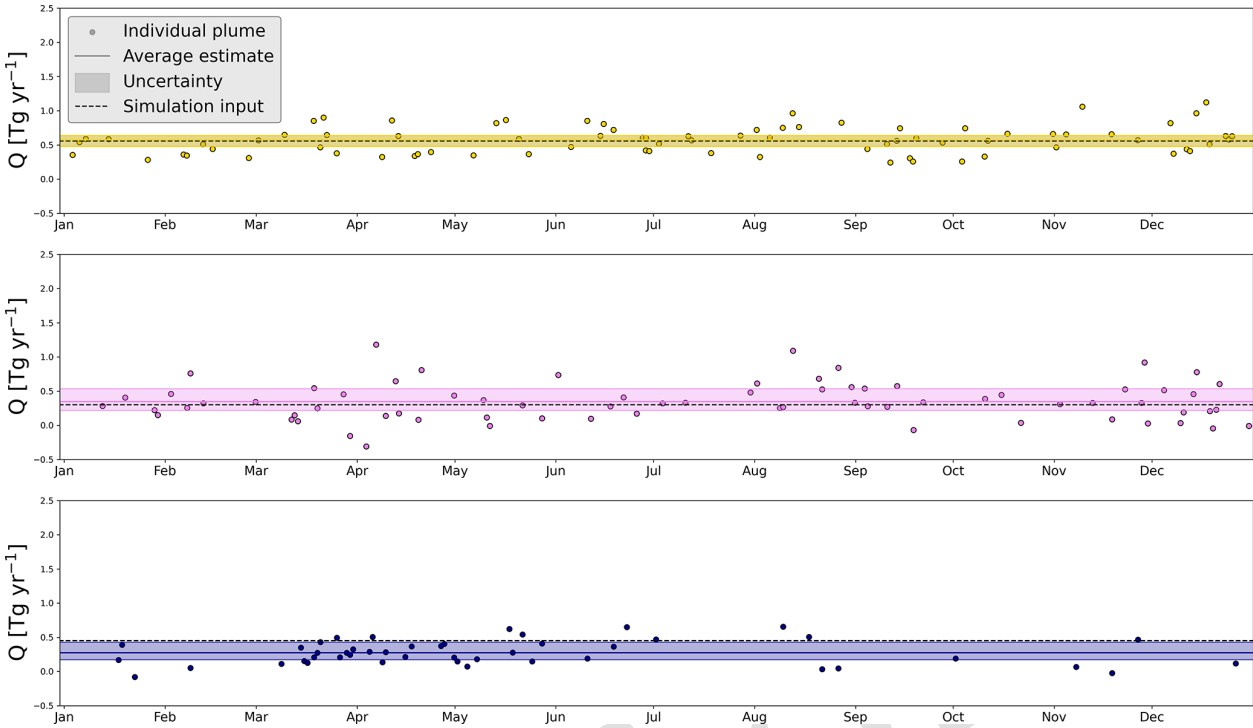

**Figure 6.** To check the validity of the CSF method for quantification of city emissions, we apply the method to simulated plumes sampled as TROPOMI would see them. The dots show CSF emission estimates for individual days over Cairo, Bamako, and Lagos respectively. The dark-colored line shows the annual CSF mean, with the uncertainty based on the emission ensemble shown by the shaded area. The simulation emission input, dotted black line, lies within the uncertainty of the mean CSF emission estimate for Cairo and Bamako, showing that the CSF can successfully quantify these urban emissions. For Lagos the emissions are underestimated, as the NCEP winds used to drive the simulation are much higher than both the GEOS-FP and ERA5 wind products. TS1

## 3.1 Northern Africa

Two cities that stand out in Fig. 7 are Algiers and Casablanca. Unlike DACCIWA, EDGAR does not include any major emission sources around these cities, even though they both have populations of 4.2 million. EDGAR also appears to largely underestimate the emissions of the two Egyptian cities that were investigated, Cairo and Alexandria, while the DACCIWA emissions for these cities agree well with our TROPOMI estimates. As we can not directly obtain the underlying emission factors and activity data that are used in EDGAR and DACCIWA, we compare the TROPOMI CO emission rates to the corresponding $CO_2$ emission rates in EDGAR, as shown in Fig. 8. The $CO_2$ emission rates are also included in Table D2 in Appendix D. Cairo, Alexandria, Casablanca, and Algiers clearly deviate from the other cities. Their much higher values for $CO_{TROPOMI}/CO_{EDGAR}$ correspond to lower values for $CO/CO_2$ in EDGAR, indicating that not the activity data but the CO emission factors for these cities are underestimated in EDGAR. This is further confirmed by the higher $CO/CO_2$ values in DACCIWA and the fact that the absolute $CO_2$ emission rates for these cities agree well between the two inventories. The underestimation in CO may point at an overestimated combustion efficiency

used in the compilation of the EDGAR emissions for this region. Similar observations over Cairo were made by MacDonald et al. (2023) when comparing measured CO and $CO_2$ concentrations.

## 3.2 South Africa

In South Africa we find closer agreement between EDGAR and TROPOMI than in northern Africa. However, the emission rates for the four considered cities in DACCIWA are on average 2.4 times higher than those based on TROPOMI (Fig. 7). The emission ratios from Fig. 8 show that the South African cities stand out from the other cities in EDGAR, as they have relatively low $CO/CO_2$ emission ratios, suggesting high average combustion efficiencies. This does not hold for DACCIWA, where the South African cities have $CO/CO_2$ emission ratios comparable to other cities. This indicates that the CO emission factors for South Africa are overestimated in DACCIWA, and these cities have higher combustion efficiencies more in line with EDGAR.

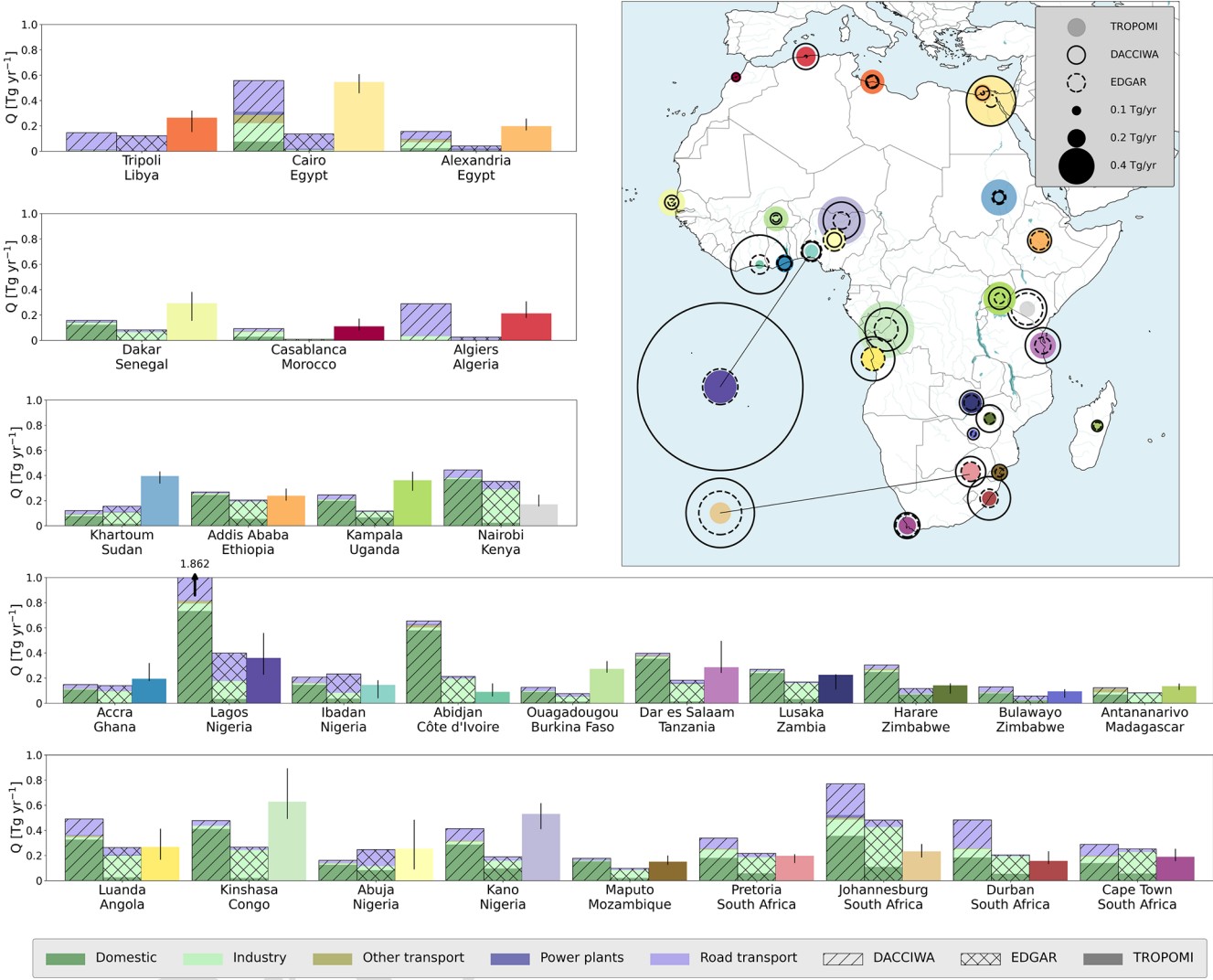

**Figure 7.** CSF emission quantifications for the largest African cities. Comparison between TROPOMI emission estimates averaged for 2019–2021 (shown as colored circles) and the DACCIWA and EDGAR v5 emission inventories for 2015 shown by the black (dashed) rings. The emission strength is indicated by the size of the circles or rings. The same comparison is made in bar plots, where the first two bars show the emission rates from DACCIWA and EDGAR respectively including the sectoral breakdown. The third bar gives the corresponding TROPOMI estimate, where the uncertainty is given by the range of the ensemble. The cities are ordered by geographical location. The emission estimate for Lagos in DACCIWA extends beyond the figure boundary.

## 3.3 Nigeria

The four investigated cities in Nigeria show varying results when comparing TROPOMI to the inventories, but the two cities that stand out are Lagos and Kano. In Lagos we estimate emissions of 0.36 (0.23–0.56) $\mathrm{Tg\,yr^{-1}}$ that are consistent with EDGAR, but DACCIWA has emissions that are 5.2 times higher, a difference which is much larger than the uncertainty in wind data discussed in Sect. 2.6. For Kano, in contrast to Lagos, we observe an emission rate of 0.53 (0.41–0.62) $\mathrm{Tg\,yr^{-1}}$, which is consistent with DACCIWA but more than twice the EDGAR estimate (0.19 $\mathrm{Tg\,yr^{-1}}$). The $CO/CO_2$ ratios of the inventories agree within 50 %, but

the differences are caused by the activity data. Figure 9 shows that the $CO_2$ emission rates in DACCIWA for Lagos, Kano, and Ibadan are respectively 8.1, 3.8, and 3.3 times higher than in EDGAR; these data are also available in Appendix D in Table D2. Comparing Nigeria's national $CO_2$ budget, there is a 24 % difference between the inventories (EDGAR 530 $\mathrm{Tg\,yr^{-1}}$ to DACCIWA 700 $\mathrm{Tg\,yr^{-1}}$), but the larger regional discrepancies (over 700 % for Lagos' $CO_2$ emissions) suggest differences in spatial allocation as well.

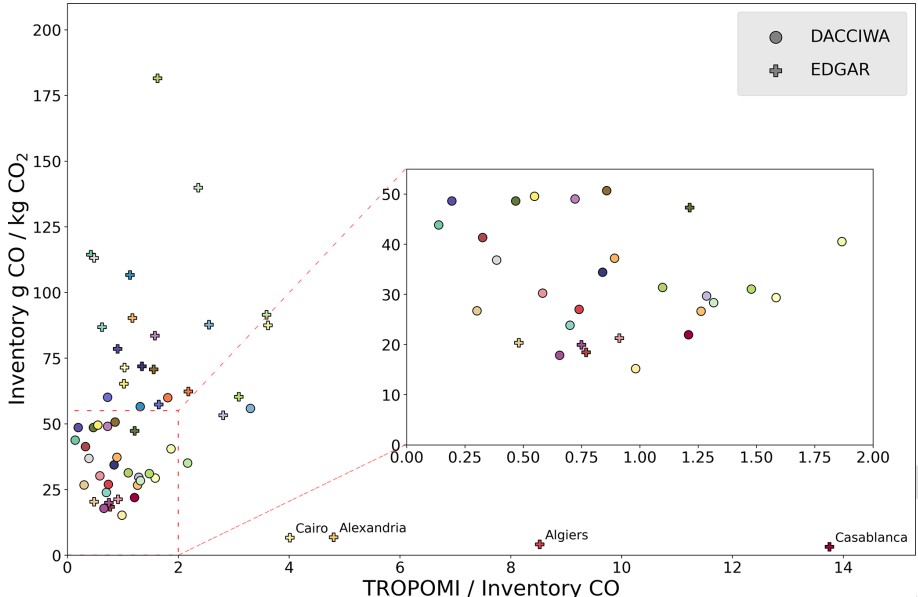

**Figure 8.** Comparison between inventory combustion completeness and TROPOMI to inventory CO estimates. Each marker represents a single city. The $CO_2$ values for both inventories include both fossil fuel and biofuel combustion emissions. As power plants hardly emit any CO per kilogram of emitted $CO_2$ due to their high combustion efficiency, the contributions of this sector are removed from the $CO_2$ values. Cairo, Alexandria, Algiers, and Casablanca have very low CO emission rates in EDGAR compared to TROPOMI and compared to EDGAR $CO_2$ emissions, which indicates that EDGAR largely overestimates the combustion efficiency for these cities. The four cities in EDGAR with $CO/CO_2$ values around 20 are all cities in South Africa, showing lower CO emission rates than the other African cities.

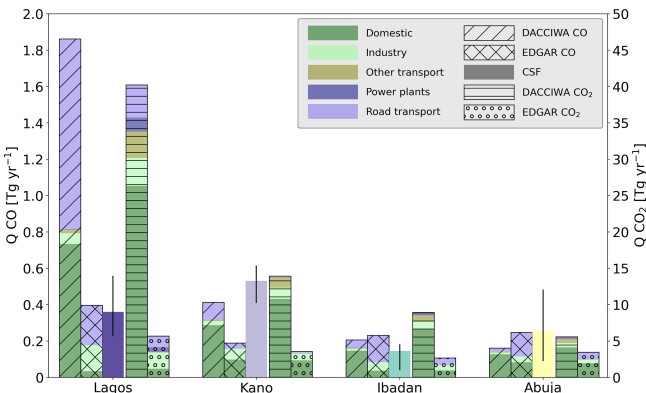

**Figure 9.** Comparison between the TROPOMI CO emission estimates and EDGAR and DACCIWA CO and $CO_2$ for four cities in Nigeria with the same color scheme as Fig. 7. The differences between the two inventories in $CO_2$ emission rates indicate a different spatial allocation – based on gridded activity data – of the national totals.

### 3.4 Côte d'Ivoire

Abidjan in Côte d'Ivoire has the largest relative discrepancy between DACCIWA ($0.65 \, \text{Tg yr}^{-1}$) and TROPOMI ($0.1 \, (0.05–0.16) \, \text{Tg yr}^{-1}$). In DACCIWA, the domestic sector contributes to 89 % of the city's emissions, and Abidjan is the city with one of the highest $CO/CO_2$ values of all investigated cities. In EDGAR, the domestic $CO/CO_2$ ra-

tio for Abidjan is 4 times lower, which would indicate a 4 times lower emission rate. This would bring the DACCIWA emission rate much closer to the TROPOMI observed emissions, indicating that, similar to South Africa, DACCIWA may overestimate the city's domestic sector's CO emission factor.

### 3.5 Libya

Tripoli, the capital of Libya, stands out as its CO emissions in both inventories are almost exclusively 90+% due to road transport. The TROPOMI estimate for this city of $0.26 \, (0.15–0.32) \, \text{Tg yr}^{-1}$ is 1.7 and 2.2 times higher than DACCIWA and EDGAR respectively. The difference can be partly explained by considering the domestic and industry sectors. In both emission inventories, the $CO/CO_2$ ratios for these sectors are 4 to 5 times lower than the mean of the other cities and 2 to 3 times lower than the next lowest city (excluding Egypt, Morocco, and Algeria). This implies that these sectors in Tripoli have high combustion efficiency compared to other cities. However, based on the TROPOMI estimate, both inventories seem to underestimate the emission factor for Tripoli, specifically for the non-road transport sectors.

### 3.6 Temporal emission patterns

With the 3-year TROPOMI dataset, we can also investigate the temporal variability of emissions. Earlier studies,

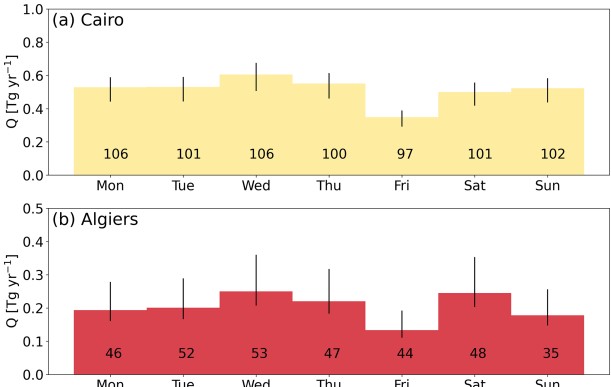

**Figure 10.** TROPOMI emission estimates over Cairo (**a**) and Algiers (**b**) for different days of the week averaged over 2019–2021. The numbers in the bars show the number of days the averages are based on. Both cities show lower emissions on Friday, consistent with road transport being the main contributor to urban emissions and Friday being the standard day off in Islamic countries.

focusing on concentration trends rather than emission estimates, have found that CO concentrations over Cairo are lower on Fridays, which is the day off in the Islamic world (Rey-Pommier et al., 2022). This "weekend effect" has also been observed for nitrogen dioxide ($NO_2$) and ozone ($O_3$), which like CO in Cairo are dominated by transport emissions (Beirle et al., 2003; Khoder, 2009; Stavrakou et al., 2020). Combined with the fact that both emission inventories agree on road transport as the main contributor to emissions in Cairo, lower CO emissions are indeed expected on Fridays when there is less commuter traffic. Figure 10 indeed shows a 32 % drop in emissions on Fridays over Cairo. A similar reduction in emissions can be seen over Algiers, which can also be attributed to reduced road traffic. Similar significant patterns were not seen for the other cities that tend to have relatively lower contributions from road traffic.

## 4  Conclusions

We adapted and calibrated the computationally efficient cross-sectional flux (CSF) method to quantify urban carbon monoxide emission rates from major cities in Africa using TROPOMI data. We determined optimal values for the parameters of the CSF by applying the method to a full year of simulated WRF plumes over three distinctly different African cities (Cairo, Lagos, and Bamako), such that the transects drawn best match the shape and curvature of the simulated plumes. These simulations were also used to calibrate the CSF's effective wind speed relationship for TROPOMI data. By applying the calibrated CSF to the simulated data with known emission rates, we found that we can quantify urban CO emissions down to $0.1\,\mathrm{Tg\,yr^{-1}}$ within 30 % uncertainty. After calibration, we applied our CSF method to TROPOMI observations of 29 of Africa's most populated

and/or emitting cities. We focus on Africa as there are relatively few dedicated emission inventories for the continent, and large uncertainties in emission rates are expected.

We compared our TROPOMI-based emission estimates with the global EDGAR emission inventory and the Africa-focused DACCIWA inventory. There are substantial differences between urban CO emissions from both inventories. We did not find vastly better average agreement of either inventory with TROPOMI. DACCIWA is closer to the TROPOMI estimate for 16 out of 29 cities. For 10 cities, the DACCIWA and TROPOMI estimates agree within the uncertainty of the TROPOMI-based estimate, but there are also cities with large significant differences of over 600 %. Compared to EDGAR, we find that nine cities agree within the uncertainty and we similarly find cities with large discrepancies.

We then evaluated our results for different regions. In northern Africa, TROPOMI observes higher emission rates than shown in EDGAR for cities in Egypt, Algeria, and Morocco. The EDGAR CO to $CO_2$ emission ratios for these four cities are relatively low, implying that the mismatch with TROPOMI may originate from the emission factors, which implies that EDGAR would overestimate the average combustion efficiency in these cities. In South Africa, the TROPOMI estimates agree with EDGAR, but the DACCIWA estimates are high in comparison. EDGAR shows lower CO to $CO_2$ ratios and hence higher combustion efficiency for South Africa compared to other parts of Africa, implying that those ratios may be too high in DACCIWA. Similarly, DACCIWA appears to underestimate the combustion efficiency in Abidjan (Côte d'Ivoire). For Tripoli (Libya), both inventories estimate lower emission rates than the estimate based on TROPOMI. Specifically the domestic and industry sectors show particularly high combustion efficiencies compared to other cities in both inventories, which can explain part of the discrepancy.

We also found some discrepancies that can be attributed to the activity data used by the inventories. We found a factor ∼ 4 lower emissions based on TROPOMI for Lagos (Nigeria) than estimated by DACCIWA. The associated DACCIWA $CO_2$ emissions are 8 times larger than in EDGAR, which can partly be a difference in activity data but also suggests a mismatch related to the spatial distribution of emissions. For Kano (Nigeria), DACCIWA estimates $CO_2$ emissions that are 4 times larger than EDGAR. Here, the TROPOMI estimate agrees better with DACCIWA, and the activity rate, which corresponds to $CO_2$ emission, in EDGAR seems to be an underestimation.

The large TROPOMI data volume enables the identification of temporal emission patterns. Over Cairo and Algiers we find significantly lower emission rates on Fridays – the local rest day – compared to other days of the week. The ability to recognize such patterns builds confidence and shows the strength of TROPOMI's daily global coverage combined

with a computationally efficient method like the CSF method developed here.

## Appendix A: TROPOMI-based CO uncertainty

The CSF method as applied in this work has many parameters that were calibrated on simulated plumes. In order to determine the uncertainty of the estimated emission rate, we have created an ensemble of emission estimates by varying these parameters. The members of the ensemble and the ranges over which they were varied are shown in Table A1. The wind databases are the three wind products as described in Sect. 2.5 and are responsible for a mean uncertainty of $-19\%$ and $+12\%$. The standard deviation threshold for spline pixels is the number of standard deviations a pixel has to be above the background concentration in order to be considered part of the plume. Pixels identified as part of the plume are only used for fitting the spline shape. The number of cross sections is the total number of transects drawn perpendicular to the direction of the plume. As described in Sect. 2.4, not all transects are taken into account in the quantification. The number of cross sections is mostly a measure for the line density or the distance between consecutive transects, as the transects are evenly spaced over the full length of the spline. The minimum pixel coverage is the minimum fraction of a line that needs to be covered by pixels and is a balance between retaining enough days with a valid estimate and not underestimating emissions. A lower limit of 50 % coverage per line will retain a lot of lines and, thus, more days with an estimate; however, the cross sections will potentially miss large parts of the plume. The distance of the background box is important, as one would like the box to be close to the city to get a good representation of the local background. However, it must not overlap with any urban emissions to have a clean background. As explained in Sect. 2.4, the transects start upwind of the city center to capture the full city plume. We vary the distance between the first transect and the city center for our uncertainty estimate. As a last member of the ensemble, we use the spread in emission estimates of the individual transects. We include the means of the transects with the lowest and highest 50 % emission rates in the ensemble.

**Table A1.** Variables used in the uncertainty analysis and the ranges over which they were varied. The resulting ensemble spreads are reported as uncertainty.

| Parameter | Domain | Default |
|---|---|---|
| Wind database | GEOS-FP 10m, ERA5 10m, and GEOS-FP PBL | GEOS-FP 10m |
| Standard deviation threshold for spline pixels | {1.2–2.4} | 1.8 |
| Number of transects | {15–25} | 20 |
| Minimum pixel coverage per transect | {50 %–90 %} | 70 % |
| Distance of background box | {0.2–0.4°} | 0.3° |
| Varying upwind distance first transect | {0–0.2°} | 0.1° |
| Transects used for estimate | {Lowest 50 %–highest 50 %} | All |

## Appendix B: TROPOMI data filtering

Although the CSF has been shown to reproduce simulated emission rates (Fig. 5), it can not be applied to every single overpass of TROPOMI. For example, days with a lot of missing pixels in the TROPOMI data can lead to underestimation of the emission rates. To prevent a positive bias, it is important to not only use days with strong, clearly visible plumes. With the filters chosen in this work, over 400 d are accepted as quantifiable over the 3-year period studied for most non-coastal cities and coastal cities with predominantly inland winds (e.g., Cairo has 713 estimates, Johannesburg 427, and Khartoum 570). With TROPOMI's daily overpasses this means that we estimate emissions on roughly 40 % of all days. Regions with fewer estimates tend to be coastal. For example, we only have estimates for 160 d over Lagos and 113 for Dakar because of limited TROPOMI coverage over water. An additional reason for a small number of valid estimates lies in the occurrence of open fires; for example, 224 orbits (42 %) are removed from our estimate over Lusaka (Zambia) due to fires within 1.5° of the city center and stronger fires within 0.75°.

The filters employed are shown in Table B1.

**Table B1.** Filtering applied to the data to ensure correct application of the CSF method.

| Description | | Value |
|---|---|---|
| Per pixel | Quality flag TROPOMI (land). | $\geq 0.7$ |
| | Quality flag TROPOMI (water). | $= 0.7$ |
| Per transect | Misalignment between plume and wind direction. | $< 45°$ |
| | Minimum pixel coverage. | $> 70\,\%$ |
| Per plume | Downwind coverage in a $0.3° \times 0.8°$ box. | $> 60\,\%$ |
| | Effective wind speed. | $> 2\,\mathrm{m\,s^{-1}}$ |
| | Maximum concentration outside the plume. | $< 200\,\mathrm{ppb}$ ($1.5°$ radius) |
| | Number of transects used for the estimate. | $> 3$ |
| | Fraction estimate transect 8–20 to transect 3–7. High emission estimates of the far away lines tend to indicate interference of different sources. | $< 2.5\times$ |
| | Second derivative of spline scaled to $1°$ pixel size. This represents the dimensionless curvature of the fitted spline. | $< 0.05$ |
| | Mismatch between the first transect and the starting pixel of the plume. | $< 0.35°$ |
| | Fire emission from Global Fire Assimilation System (GFAS) database, Kaiser et al. (2012). | $< 23\,\mathrm{Mg\,h^{-1}}$ ($0.75°$ radius), $< 57\,\mathrm{Mg\,h^{-1}}$ ($1.5°$ radius) |

## Appendix C: CSF estimates

**Table C1.** Emission estimates for the studied African cities by applying the CSF method to the TROPOMI CO product (2019–2021), as well as the corresponding emission rates according to the DACCIWA (2015) and EDGAR (2015) inventories. All emission rates are in teragrams per year (Tg yr$^{-1}$). Population is taken from the Center for International Earth Science Information Network (CIESIN, 2018).

| City | Country | Population | TROPOMI estimate | Lower limit | Upper limit | DACCIWA | EDGAR |
|------|---------|-----------|------------------|-------------|-------------|---------|-------|
| Algiers | Algeria | 4.2 M | 0.213 | 0.177 | 0.307 | 0.288 | 0.025 |
| Luanda | Angola | 5.1 M | 0.268 | 0.166 | 0.413 | 0.489 | 0.263 |
| Ouagadougou | Burkina Faso | 2.8 M | 0.273 | 0.243 | 0.335 | 0.126 | 0.076 |
| Kinshasa | Congo | 7.4 M | 0.628 | 0.49 | 0.894 | 0.477 | 0.266 |
| Abidjan | Côte d'Ivoire | 7.1 M | 0.09 | 0.054 | 0.157 | 0.654 | 0.266 |
| Alexandria | Egypt | 3.3 M | 0.197 | 0.164 | 0.257 | 0.156 | 0.041 |
| Cairo | Egypt | 16.7 M | 0.546 | 0.456 | 0.608 | 0.556 | 0.136 |
| Addis Ababa | Ethiopia | 4.4 M | 0.239 | 0.2 | 0.295 | 0.268 | 0.204 |
| Accra | Ghana | 3.5 M | 0.194 | 0.176 | 0.318 | 0.148 | 0.172 |
| Nairobi | Kenya | 4.8 M | 0.17 | 0.153 | 0.247 | 0.441 | 0.353 |
| Tripoli | Libya | 1.2 M | 0.264 | 0.151 | 0.32 | 0.146 | 0.121 |
| Antananarivo | Madagascar | 3.0 M | 0.135 | 0.104 | 0.156 | 0.123 | 0.083 |
| Casablanca | Morocco | 4.2 M | 0.11 | 0.078 | 0.171 | 0.091 | 0.008 |
| Maputo | Mozambique | 2.5 M | 0.151 | 0.125 | 0.199 | 0.176 | 0.097 |
| Abuja | Nigeria | 2.6 M | 0.255 | 0.089 | 0.483 | 0.161 | 0.247 |
| Ibadan | Nigeria | 2.2 M | 0.145 | 0.039 | 0.183 | 0.207 | 0.232 |
| Kano | Nigeria | 5.3 M | 0.531 | 0.409 | 0.616 | 0.413 | 0.189 |
| Lagos | Nigeria | 10.9 M | 0.36 | 0.227 | 0.558 | 1.862 | 0.397 |
| Dakar | Senegal | 4.2 M | 0.293 | 0.154 | 0.381 | 0.157 | 0.081 |
| Cape Town | South Africa | 4.1 M | 0.189 | 0.155 | 0.252 | 0.288 | 0.252 |
| Durban | South Africa | 3.3 M | 0.157 | 0.132 | 0.233 | 0.482 | 0.204 |
| Johannesburg | South Africa | 9.1 M | 0.232 | 0.184 | 0.291 | 0.77 | 0.482 |
| Pretoria | South Africa | 6.4 M | 0.197 | 0.139 | 0.209 | 0.338 | 0.216 |
| Khartoum | Sudan | 3.1 M | 0.396 | 0.336 | 0.432 | 0.12 | 0.155 |
| Dar es Salaam | Tanzania | 5.4 M | 0.286 | 0.241 | 0.496 | 0.396 | 0.181 |
| Kampala | Uganda | 4.3 M | 0.362 | 0.279 | 0.431 | 0.245 | 0.117 |
| Lusaka | Zambia | 2.4 M | 0.226 | 0.108 | 0.23 | 0.269 | 0.168 |
| Bulawayo | Zimbabwe | 0.8 M | 0.094 | 0.043 | 0.111 | 0.13 | 0.057 |
| Harare | Zimbabwe | 2.7 M | 0.142 | 0.076 | 0.157 | 0.304 | 0.117 |

## Appendix D: Inventory emission rates

**Table D1.** Sectoral breakdown of the CO emission rates for the studied African cities according to the DACCIWA and EDGAR inventory. All emission rates are in gigagrams per year (Gg yr$^{-1}$).

| City | DACCIWA | | | | | EDGAR | | | | |
|------|---------|---|---|---|---|-------|---|---|---|---|
| | Domestic | Industry | Other | Power | Road | Domestic | Industry | Other | Power | Road |
| Algiers | 0.3 | 33.7 | 0.6 | 0.9 | 252.2 | 1.3 | 2.7 | 0.2 | 0.4 | 20.0 |
| Luanda | 326.1 | 19.4 | 12.0 | 3.4 | 129.0 | 25.4 | 174.0 | 0.3 | 6.5 | 56.1 |
| Ouagadougou | 90.3 | 8.0 | 0.0 | 0.8 | 26.4 | 13.2 | 39.8 | 0.1 | 2.8 | 20.0 |
| Kinshasa | 411.1 | 25.4 | 0.0 | 0.0 | 40.4 | 20.1 | 222.3 | 0.1 | 0.1 | 23.5 |
| Abidjan | 579.9 | 21.4 | 20.4 | 7.0 | 25.5 | 10.8 | 183.6 | 0.3 | 1.6 | 15.7 |
| Alexandria | 24.3 | 46.7 | 20.7 | 4.4 | 59.6 | 3.4 | 1.8 | 0.3 | 15.8 | 19.9 |
| Cairo | 75.8 | 145.5 | 64.6 | 26.1 | 244.3 | 3.5 | 6.4 | 1.1 | 9.3 | 115.9 |
| Addis Ababa | 245.3 | 8.5 | 1.8 | 0.0 | 12.3 | 55.3 | 134.8 | 0.2 | 0.0 | 13.9 |
| Accra | 108.2 | 3.2 | 3.9 | 0.0 | 32.4 | 5.0 | 90.2 | 1.0 | 0.5 | 43.2 |
| Nairobi | 370.8 | 11.7 | 0.2 | 2.3 | 59.1 | 23.5 | 266.9 | 0.3 | 2.1 | 60.0 |
| Tripoli | 3.1 | 4.2 | 0.0 | 6.6 | 131.6 | 1.9 | 0.1 | 0.1 | 0.0 | 118.9 |
| Antananarivo | 68.8 | 17.0 | 30.2 | 0.1 | 6.5 | 7.5 | 68.3 | 0.1 | 0.0 | 7.3 |
| Casablanca | 29.2 | 37.1 | 4.6 | 0.5 | 20.0 | 1.5 | 3.9 | 0.0 | 0.2 | 2.2 |
| Maputo | 153.1 | 5.3 | 0.1 | 3.1 | 14.7 | 21.0 | 62.7 | 0.0 | 0.3 | 13.1 |
| Abuja | 126.2 | 10.2 | 3.6 | 0.1 | 21.0 | 82.6 | 32.1 | 0.0 | 0.0 | 131.9 |
| Ibadan | 145.8 | 11.7 | 4.1 | 0.3 | 44.7 | 36.2 | 46.4 | 0.0 | 0.0 | 149.1 |
| Kano | 286.3 | 23.0 | 8.1 | 0.0 | 95.5 | 96.9 | 60.6 | 0.0 | 0.0 | 31.0 |
| Lagos | 733.5 | 59.0 | 20.8 | 2.4 | 1046.3 | 33.2 | 143.5 | 0.1 | 0.7 | 219.8 |
| Dakar | 123.3 | 17.4 | 0.1 | 6.1 | 10.0 | 2.7 | 62.7 | 0.2 | 3.2 | 12.6 |
| Cape Town | 139.9 | 50.7 | 5.7 | 1.4 | 90.5 | 56.9 | 170.1 | 0.7 | 0.0 | 23.8 |
| Durban | 183.8 | 66.6 | 7.5 | 0.0 | 224.4 | 53.3 | 139.4 | 0.3 | 0.0 | 10.9 |
| Johannesburg | 355.8 | 128.9 | 14.5 | 22.4 | 247.9 | 108.8 | 317.5 | 0.1 | 0.0 | 55.1 |
| Pretoria | 180.4 | 65.4 | 7.4 | 0.9 | 83.4 | 57.3 | 129.0 | 0.1 | 0.4 | 29.3 |
| Khartoum | 77.2 | 10.8 | 3.8 | 1.6 | 26.2 | 16.3 | 90.0 | 0.1 | 5.6 | 42.6 |
| Dar es Salaam | 354.2 | 15.0 | 9.6 | 1.7 | 15.8 | 15.1 | 138.1 | 0.1 | 10.4 | 17.7 |
| Kampala | 198.8 | 10.5 | 0.1 | 1.2 | 34.2 | 65.6 | 42.5 | 0.0 | 0.0 | 8.8 |
| Lusaka | 239.0 | 11.7 | 1.2 | 0.0 | 16.7 | 31.7 | 126.3 | 0.0 | 0.0 | 9.8 |
| Bulawayo | 79.1 | 4.3 | 3.5 | 0.2 | 42.4 | 21.5 | 2.0 | 0.0 | 0.2 | 33.0 |
| Harare | 248.2 | 13.3 | 10.8 | 0.2 | 31.5 | 69.8 | 4.5 | 0.1 | 0.4 | 41.7 |

**Table D2.** Sectoral breakdown of the $CO_2$ emission rates for the studied African cities according to the DACCIWA and EDGAR inventory. All emission rates are in teragrams per year ($Tg\,yr^{-1}$).

| City | DACCIWA | | | | | EDGAR | | | | |
|---|---|---|---|---|---|---|---|---|---|---|
| | Domestic | Industry | Other | Power | Road | Domestic | Industry | Other | Power | Road |
| Algiers | 2.0526 | 2.3547 | 0.3026 | 0.7006 | 5.9577 | 2.2141 | 2.3123 | 0.1231 | 0.0446 | 1.4081 |
| Luanda | 4.8853 | 1.4104 | 2.2871 | 0.0 | 1.2816 | 1.8331 | 1.4681 | 0.1276 | 1.6398 | 0.6036 |
| Ouagadougou | 2.1571 | 0.5752 | 0.701 | 0.4603 | 0.1553 | 0.3469 | 0.2106 | 0.1102 | 0.4868 | 0.1633 |
| Kinshasa | 9.1058 | 2.4811 | 4.9228 | 0.0 | 0.3205 | 0.4757 | 1.1183 | 0.0897 | 0.0245 | 0.2184 |
| Abidjan | 5.6168 | 1.2141 | 7.6905 | 4.1632 | 0.3916 | 0.3994 | 1.1479 | 0.1115 | 1.1956 | 0.1948 |
| Alexandria | 1.7076 | 3.1371 | 0.3293 | 0.0 | 0.6755 | 0.8145 | 4.1211 | 0.0639 | 23.3842 | 0.9927 |
| Cairo | 9.7364 | 17.8872 | 3.5146 | 17.7906 | 5.4195 | 3.1328 | 11.4975 | 0.7648 | 13.7938 | 4.9941 |
| Addis Ababa | 5.2452 | 0.4889 | 1.0817 | 0.0 | 0.3851 | 1.2733 | 0.5438 | 0.2141 | 0.0005 | 0.2297 |
| Accra | 1.172 | 0.3122 | 0.7803 | 0.6024 | 0.3533 | 0.231 | 0.6034 | 0.4059 | 0.3562 | 0.3729 |
| Nairobi | 5.3718 | 0.9503 | 4.7633 | 0.429 | 0.8817 | 0.6676 | 1.6001 | 0.2878 | 0.4414 | 0.5657 |
| Tripoli | 0.3477 | 0.8711 | 0.3068 | 7.4683 | 0.9077 | 0.252 | 0.6718 | 0.022 | 0.0 | 0.9969 |
| Antananarivo | 1.7812 | 0.9208 | 1.0695 | 0.0798 | 0.1474 | 0.205 | 0.144 | 0.046 | 0.0195 | 0.0622 |
| Casablanca | 1.4466 | 2.1949 | 0.0158 | 4.0377 | 0.4885 | 0.7635 | 1.2911 | 0.0514 | 1.2078 | 0.3914 |
| Maputo | 2.3379 | 0.2568 | 0.6895 | 0.0 | 0.1877 | 0.4888 | 0.7266 | 0.0355 | 0.0795 | 0.1202 |
| Abuja | 4.1275 | 0.6013 | 0.5368 | 0.1024 | 0.2142 | 1.9347 | 0.4962 | 0.0413 | 0.0 | 0.9858 |
| Ibadan | 6.678 | 0.9728 | 0.8684 | 0.2404 | 0.1572 | 0.944 | 0.7214 | 0.0006 | 0.0006 | 1.0063 |
| Kano | 10.773 | 1.5693 | 1.401 | 0.0311 | 0.1601 | 2.3444 | 0.9346 | 0.0206 | 0.0008 | 0.2502 |
| Lagos | 26.3529 | 3.8388 | 3.5988 | 1.9078 | 4.5136 | 1.1902 | 2.2007 | 0.0889 | 0.6383 | 1.5765 |
| Dakar | 1.5405 | 1.01 | 0.9282 | 1.9502 | 0.3954 | 0.1379 | 0.4475 | 0.1115 | 1.2811 | 0.2292 |
| Cape Town | 3.1332 | 10.8663 | 1.2788 | 1.0896 | 0.823 | 1.759 | 9.0686 | 0.247 | 0.0009 | 1.5695 |
| Durban | 2.3207 | 8.0484 | 0.543 | 0.1116 | 0.7436 | 1.5599 | 8.5055 | 0.0943 | 0.1335 | 0.9045 |
| Johannesburg | 5.5358 | 19.1984 | 2.2775 | 23.3095 | 1.7699 | 3.1791 | 17.0838 | 0.1623 | 4.9948 | 3.2495 |
| Pretoria | 2.2686 | 7.8676 | 0.5308 | 3.0576 | 0.5046 | 1.6006 | 6.6839 | 0.1254 | 3.4728 | 1.7434 |
| Khartoum | 0.6315 | 0.9569 | 0.3614 | 0.1673 | 0.198 | 0.5049 | 0.6555 | 0.101 | 1.4848 | 0.5062 |
| Dar es Salaam | 5.582 | 1.0448 | 1.4222 | 1.073 | 0.0259 | 0.3639 | 1.5205 | 0.0908 | 1.3461 | 0.1923 |
| Kampala | 5.0997 | 0.8774 | 1.6523 | 0.5054 | 0.2586 | 1.4997 | 0.329 | 0.009 | 0.0 | 0.1042 |
| Lusaka | 3.8478 | 1.6338 | 2.2394 | 0.0 | 0.0951 | 0.7361 | 1.4979 | 0.0327 | 0.0 | 0.0699 |
| Bulawayo | 1.8723 | 0.1759 | 0.0062 | 0.2274 | 0.1068 | 0.5132 | 0.2041 | 0.0134 | 0.2151 | 0.263 |
| Harare | 5.522 | 0.5187 | 0.0183 | 0.2416 | 0.193 | 1.6508 | 0.3461 | 0.059 | 0.4645 | 0.4179 |

**Code and data availability.** TROPOMI CO data (https://doi.org/10.5270/S5P-bj3nry0, Copernicus Sentinel-5P, 2021) are publicly available from the ESA Sentinel-5P data hub at https://s5phub.copernicus.eu (last access: 1 June 2023). ERA5 wind data are available at https://www.ecmwf.int/en/forecasts/dataset/ecmwf-reanalysis-v5 (Hersbach et al., 2022). WRF-Chem code is available at https://github.com/wrf-model/WRF/releases (Powers et al., 2023); in this work, version 4.1.5 was used. EDGAR v5 CO data are available at https://edgar.jrc.ec.europa.eu/dataset_ap50 (Crippa et al., 2021). EDGAR v5 $CO_2$ data are available at https://edgar.jrc.ec.europa.eu/dataset_ghg50 (Crippa et al., 2022). DACCIWA CO and $CO_2$ data are available at https://doi.org/10.25326/56 (Keita et al., 2020) at the Emissions of atmospheric Compounds and Compilation of Ancillary Data (ECCAD) system. GPW v4 gridded population density is available at https://doi.org/10.7927/H49C6VHW (CIESIN, 2018). Open fire emissions from GFAS are available at https://atmosphere.copernicus.eu/global-fire-emissions (Kaiser et al., 2022).

**Author contributions.** GL, JDM, and IA designed the study. GL performed the TROPOMI analysis with contributions from JDM, HACDvdG, AS, and IA. GL and JDM wrote the paper with contributions from all authors. JDM, HACDvdG, and AS contributed to the comparison between TROPOMI and emission inventories. TB provided the TROPOMI carbon monoxide data and associated support.

**Competing interests.** The contact author has declared that none of the authors has any competing interests.

**Disclaimer.** Publisher's note: Copernicus Publications remains neutral with regard to jurisdictional claims in published maps and institutional affiliations.

**Acknowledgements.** We thank the team that realized the TROPOMI instrument and its data products, consisting of the part-

nership between Airbus Defence and Space Netherlands, KNMI, SRON, and TNO, commissioned by NSO and ESA. Sentinel-5 Precursor is part of the EU Copernicus program, and Copernicus Sentinel-5P data (2019–2021) have been used. We thank SURF (https://www.surf.nl/, last access: 1 June 2023) for the support in using the National Supercomputer Snellius. We thank Sekou Keita and Cathy Liousse (CNRS, Toulouse) for development and early access to the DACCIWA_v2 dataset.

**Financial support.** This research has been supported by the European Commission Horizon 2020 Framework Programme (CoCO2 project, grant no. 958927).

**Review statement.** This paper was edited by Bryan N. Duncan and reviewed by two anonymous referees.

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

## Remarks from the typesetter