# Peer review of "Quantification of carbon monoxide emissions from African cities using TROPOMI"

_Atmospheric Chemistry and Physics, 2023_

## Referee Comment (RC1)

**Review of "Quantification of carbon monoxide emissions from African cities using TROPOMI" by G. Leguijt et al. submitted to ACP, 2023**

**General Description and Recommendation:**

The authors quantify urban CO emissions in large cities in Africa by applying the cross-sectional flux (CSF) method to satellite observations of CO from TROPOMI following theoretical assessment of the approach using synthetic columns from the WRF-Chem model. This is a potentially interesting application of this technique, but the paper in its current form is problematic, as there is limited reference to past studies focusing on Africa in the introduction, the methods are difficult to follow making it challenging to review the results, and absent is an assessment of contamination of urban CO from widespread and intense open burning of biomass. My recommendation is to resubmit the manuscript following major revision.

The introduction only really includes information about dominant emission sources from a global study, rather than using information that has been gained from regional emission inventories (DACCIWA, DICE-Africa) and local field campaign measurements of emissions or concentration measurements that provide constraints on emissions.

The methods provide insufficient or unclear information to follow what was done. My concerns are given below by subsections:

Section 2.3: It's not clear how the model is sampled (during the satellite overpass time?) to obtain synthetic columns.

Section 2.4: What is the quantification point that is referred to a few times in this section? A width and length are given, but why not longitude and latitude (L128)? Why use DACCIWA to identify the city centre, when Google Maps could be used? From "-90 to 90 deg" (L129) suggests the box ends up back where it started. Why is the transect (L143) 0.1 degree longer than the initial box width given in line 141? What is "the plume mask" (L144), as it's not defined earlier? Is the "3 pixels" (L145) for each transect? Is the mean of overlapping pixels assessed for the "two consecutive lines" (L147)? Given that the CO pixels in Figure 2(c) are not a plume, should scenes such as this one really be processed and used to estimate urban CO emissions? Where is the city centre located in Figure 2? What purpose does the wind speed colorbar serve in Figure 2? Is "wind speed at each transect" (L152) from GEOS-FP and, if so, what is the spatial resolution? What's the effect of removing (L156) the "first two lines" on the emissions estimates?

Section 2.6: It would be helpful in the first sentence to state the purpose of this Calibration to make clearer why this is done. What "TROPOMI filtering" (L170, L186) is this referring to? It would be helpful to give some context to "injection heights" in L177 by indicating what range is expected for a city plume. What's the relevance of being able to scale modelled emissions up or down (L188-L190)? Where does the "emission sector" information come from (L190)? Why use GEOS-FP instead of WRF winds (L194-L195)?

Missing from the paper is an assessment of contamination of urban CO due to CO (primary and secondary) from open burning of biomass. This is a very large source of air pollution during the dry burning season in large portions of northern and southern Africa. Given this, it would improve confidence in application of the CSF approach for deriving urban CO emissions and evaluating emission inventories if it can be demonstrated that there is limited or no contamination from open burning of biomass.

---

## Author Comment (AC1)

We thank both reviewers for their comments. In addition to the main responses, we have included a reference to a recent paper showing a comparison between CO and $CO_2$ concentrations over Cairo which is in agreement with our findings.

**Review 1:**

**Review of "Quantification of carbon monoxide emissions from African cities using TROPOMI" by G. Leguijt et al. submitted to ACP, 2023**

**General Description and Recommendation:**

The authors quantify urban CO emissions in large cities in Africa by applying the cross- sectional flux (CSF) method to satellite observations of CO from TROPOMI following theoretical assessment of the approach using synthetic columns from the WRF-Chem model. This is a potentially interesting application of this technique, but the paper in its current form is problematic, as there is limited reference to past studies focusing on Africa in the introduction, the methods are difficult to follow making it challenging to review the results, and absent is an assessment of contamination of urban CO from widespread and intense open burning of biomass. My recommendation is to resubmit the manuscript following major revision.

We have improved the explanation of our method such that it is easier to reproduce. As the specific points above are also mentioned in more detail below, we will put our responses there.

> The introduction only really includes information about dominant emission sources from a global study, rather than using information that has been gained from regional emission inventories (DACCIWA, DICE-Africa) and local field campaign measurements of emissions or concentration measurements that provide constraints on emissions.

We have added information focused on Africa rather than only providing global numbers. In addition, we added information on local field measurements in Africa that indicate major emissions from both transport and residential heating.

> As transport (23%) and residential heating (35%) are key contributors to total anthropogenic CO emissions (Zhong et al., 2017), cities are an important source of CO. **In Africa, the contributions of transport and residential heating are estimated at 27% and 38% of anthropogenic CO emissions respectively in the DICE-Africa inventory (Marais and Wiedinmyer, 2016) and 17% and 72% in the DACCIWA inventory (Keita et al., 2021). The importance of these two sectors is further confirmed by a large number of ground-based measurements specifically aimed at traffic (Diab et al., 2005; Lindén et al., 2008; Zakari et al., 2020; Doumbia et al., 2021) and domestic heating (Havens et al., 2018; Kansiime et al., 2022; Saleh et al., 2023) that show CO concentrations in**

*African cities exceeding air quality guidelines by the World Health Organisation.*

The methods provide insufficient or unclear information to follow what was done. My concerns are given below by subsections:
Section 2.3: It's not clear how the model is sampled (during the satellite overpass time?) to obtain synthetic columns.

To improve the methods section as a whole, we have added a flowchart as Figure 1 and incorporated references to the flowchart in the methods section. The sampling is explained in detail later in the manuscript, in Section 2.6. In Section 2.3, we now mention the sampling is done at the TROPOMI overpass time and have made a reference to Section 2.6.

*Experiment (OSSE). An OSSE is an experiment where a model or method is applied to synthetic data to evaluate the benefit of using this data and/or method. Which for this work means evaluating whether the CSF can be used to correctly estimate emissions from TROPOMI-like synthetic data. Figure 1 shows the roles of the different data products that are used and further described in Section 2.1 to 2.6. In addition, in Section 2.6 we show that the CSF method can be successfully applied to simulated data.*

[Figure]

*Figure 1. Schematic description of the use of the different data products within the OSSE and the subsequent emission quantification using TROPOMI data. The data products are discussed in the Section 2.1-2.6. First, the CSF is applied to simulated synthetic plumes in order to determine appropriate values for the various parameters used in our method. Second, an effective wind is calibrated by using the known emission rates of the simulated plumes following the procedure by Varon et al. (2018). The CSF, now calibrated on the synthetic plumes, is subsequently applied to satellite data to estimate emission rates of African cities.*

*To maintain flexibility over model output the different sectors in the emission inventories (19 for EDGAR and 6 for DACCIWA) are simulated separately. **We sample the model output (at the TROPOMI overpass time) to facilitate comparison to the TROPOMI carbon monoxide data as discussed in detail in Section 2.6.***

Section 2.4: What is the quantification point that is referred to a few times in this section? A width and length are given, but why not longitude and latitude (L128)? Why use DACCIWA to identify the city centre, when Google Maps could be used?

As the term "quantification point" is indeed confusing, we have rephrased our explanation using the city center as reference. As the box rotates each day with the wind direction its longitude and latitude bounds vary each day while its width and length are constant. We have added an explanation why we use the emission inventory to determine the city center.

*We start by determining the **city center, for our purposes defined as the location at the center of the urban emissions and therefore best representing the origin of the city's total emission.** The **location of the** center is determined by taking the **weighted** average position of the pixels in DACCIWA **that are part of the city mask introduced in Section 2.2. Weights of the pixels are** equal to their emission rate. **By determining the city location using the emission inventory, we ensure that we are comparing similar regions when we compare our satellite-based emission estimates with the emission inventories in Section 3. To make sure the entire plume is downwind, we start the transects of our CSF 0.1° upwind of this city center.***

From "-90 to 90 deg" (L129) suggests the box ends up back where it started.

We have rephrased L129 to better describe how we optimize the wind direction based on the concentration rather than only relying on the reanalysis wind product.

*As the wind direction is an important source of uncertainty, the downwind direction can not be solely based on the GEOS-FP reanalysis wind data. Instead, following Sadavarte et al. (2021a), we infer the wind direction from the satellite observations by **selecting the direction of the highest mean downwind concentration within 90°** of the reanalysis wind direction. **To do so, we calculate** the mean **downwind** concentration **over 180 boxes (0.1° width and 0.4° length) rotated at 1° intervals, and pick the direction with the highest downwind concentration.***

Why is the transect (L143) 0.1 degree longer than the initial box width given in line 141? What is "the plume mask" (L144), as it's not defined earlier? Is the "3 pixels" (L145) for each transect? Is the mean of overlapping pixels assessed for the "two consecutive lines" (L147)?

We have added an explanation why we take the transects broader than the plume itself. We have renamed and added an explanation of the plume mask (now spline mask) to make both L144 and the "3 pixels" from L145 clearer. L147 was indeed missing a proper explanation which we have added.

> *After determining the initial direction of the plume, **we need to better capture the shape of the plume to draw transects perpendicular to the plume. The** shape of the plume is determined in two steps. First, we select all pixels in a downwind box (0.3° width, 0.8° length) that exceed the mean concentration in the surrounding 3°x3° area by more than 1.8 standard deviations, **these pixels are referred to as the spline mask**. We then fit a 2D-spline (0.8° length) through the resulting **spline** mask. If, due to a lack of signal or missing pixels, the **spline** mask contains fewer than 3 pixels, a spline fit is unlikely to capture the true plume shape **and we use a straight line in the (optimized) wind direction instead (Fig. 3C).***

> *The transects (0.4° width) are drawn perpendicular to the spline, separated by 0.04°. **The transects have a larger width than the box used to determine the spline mask to ensure the transects cover the entire plume width.** All pixels overlapping with the transects are used in the **emission** quantification.*

> *We stop drawing transects **when the emission rate estimates of** two consecutive **transects** are more than one standard deviation below the mean estimate of the earlier transects, indicating the end of the plume. **Transects** with less than 70% pixel coverage are removed from the estimate as they will not have a complete integral, resulting in underestimated emissions.*

> Given that the CO pixels in Figure 2(c) are not a plume, should scenes such as this one really be processed and used to estimate urban CO emissions?

If one would exclude the days with no clear plume -such as the one in figure 2(c)-, the final average emission rate would be systematically biased high. We have added a sentence to clarify this in the text.

> ***We also include days where no clear plume is visible to avoid systematic overestimation of the average emission rate. Using Eq. 2, an emission estimate can be derived for every transect.***

> Where is the city centre located in Figure 2? What purpose does the wind speed colorbar serve in Figure 2? Is "wind speed at each transect" (L152) from GEOS-FP and, if so, what is the spatial resolution?

By removing the earlier mentioned quantification point from the explanation of our method, we now only show the city center in each figure. We have added an explanation for the wind speed colorbar as well as an explanation of the GEOS-FP winds and their resolution. Due to a plotting error the winds shown earlier were accidentally squared, which

is why they exceeded the colorscale. This issue has been fixed and we have added the GEOS-FP winds on their native resolution to give a better representation of the product.

> *The arrows show **direction and magnitude of** GEOS-FP 10-m winds **at their native 0.25° x 0.3125° resolution** (Molod et al., 2012).*

> *We therefore use the wind speed at each transect instead of a single wind speed for the entire plume. **The wind speed at the transects is calculated in two steps. First, a wind speed is calculated for each TROPOMI pixel by interpolation of the reanalysis wind product. Second, the wind speed for each transect is determined by taking the average wind speed of the overlapping TROPOMI pixels, weighted by the length of the overlap.***

> *To quantify TROPOMI plumes over all major cities in Africa we will use the NASA/GMAO GEOS-FP wind fields (Molod et al., 2012) rather than the WRF simulated wind fields **which are available only** for the 3 cities selected for evaluation and calibration. **For each TROPOMI pixel, we spatially interpolate the GEOS-FP wind field, which has a 0.25° x 0.3125° spatial resolution and a 1-hour time resolution. The NCEP winds that drive the WRF simulated wind fields have a coarser time resolution of 6 hours.***

> What's the effect of removing (L156) the "first two lines" on the emissions estimates?

We have added an explanation of the wind speed per transect and the effect of removing the initial two transects.

> *Similar to trends observed in (Sadavarte et al., 2021b), the **first two transects** are found to have roughly 30% lower emissions than the **transects** further away, **which have a stable mean emission rate.** This pattern is consistent across the cities investigated. One reason is that the early plume only captures part of the city**'s** emissions, another explanation is that the associated pixels might see a partial-pixel absorption saturation effect (Pandey et al., 2019). **Incorporating the first two transects would result in an average underestimation of emissions by 8%.** We therefore remove the first two **transects** from the emission **estimation**.*

> Section 2.6: It would be helpful in the first sentence to state the purpose of this Calibration to make clearer why this is done.

This is a good point which was also raised by the other reviewer. We have added a flowchart at the start of section 2 to highlight the different steps in our approach, including the calibration described in this section. In addition, we have added the following explanation at the start of Section 2.6:

> ***This section describes the application of the CSF to simulated CO column mixing ratios. The simulations are used to determine parameter settings (e.g. spline length and transect width) and to calibrate an effective wind (Varon et al.,***

> *2018) for TROPOMI sized pixels. In addition, the simulations are used to evaluate how well the CSF can quantify emission rates of simulated plumes.*

What "TROPOMI filtering" (L170, L186) is this referring to?

We have rephrased the text to make the TROPOMI filtering clearer.

> *We also calculate "idealized" pressure weighted columns, **which assume a uniform vertical sensitivity (flat averaging kernel),** over the TROPOMI footprints without **taking into account whether there is a valid** TROPOMI **observation** as a first check to see whether the CSF can reproduce the emissions used as model input.*

> *After determination of the effective wind on plumes with idealized pressure profiles, we test the performance of the CSF on more realistically sampled plumes which include the TROPOMI **quality filtering and averaging kernel sensitivities as described in Section 2.1 to see whether the CSF can correctly quantify emission rates from synthetic observations with quality filtering and non-uniform vertical sensitivity.***

It would be helpful to give some context to "injection heights" in L177 by indicating what range is expected for a city plume.

We have added information on the vertical profiles of emissions in the simulations to Section 2.3.

> *We use emissions from the global Emissions Database for Global Atmospheric Research (EDGAR) version 5 and the Africa-focused Dynamics-Aerosol-Chemistry-Cloud Interactions in West-Africa (DACCIWA) inventory **distributed across the vertical model levels according to the sector specific vertical profiles provided by Bieser et al. (2011). Typical injection heights for CO emissions from transport and the residential sector are 0-20 m, while emissions from industry are typically injected into the atmosphere at 100-200 m (Bieser et al., 2011).***

What's the relevance of being able to scale modelled emissions up or down (L188-L190)? Where does the "emission sector" information come from (L190)? Why use GEOS-FP instead of WRF winds (L194-L195)?

We have added an explanation in the text to address the scaling, the emission sectors and their relevance for our method. We also added information about the different winds, which is also made clearer by the previously mentioned flowchart.

> *As the modeled output concentration from the WRF simulations without chemistry scales linearly with the magnitude of the input emissions, emissions **from the different sectors provided by the bottom-up inventories** can be scaled up and down without having to rerun the chemical transport model. **This allows us to simulate plumes from cities with different emission rates with limited***

*effect on the simulated background through scaling of the* emission sector most concentrated in the considered urban area. *We use this to determine the lower limit to which our method can be trusted.*

*To quantify TROPOMI plumes over all major cities in Africa we will use the NASA/GMAO GEOS-FP wind fields (Molod et al., 2012) rather than the WRF simulated wind fields **which are available only** for the 3 cities selected for evaluation and calibration. **For each TROPOMI pixel, we spatially interpolate the GEOS-FP wind field, which has a $0.25°$ x $0.3125°$ spatial resolution and a 1-hour time resolution. The NCEP winds that drive the WRF simulated wind fields have a coarser time resolution of 6 hours.***

*For Lagos the emission rate is underestimated when using the WRF simulations as the NCEP wind fields that drive the simulations are higher than **both** the GEOS-FP and ERA5 wind products specifically over Lagos **by about 60%. The difference between the wind products might be caused by the fact that Lagos lies in the West-African monsoon region where transport has been shown to be difficult to model (Liu et al., 2014).***

Missing from the paper is an assessment of contamination of urban CO due to CO (primary and secondary) from open burning of biomass. This is a very large source of air pollution during the dry burning season in large portions of northern and southern Africa. Given this, it would improve confidence in application of the CSF approach for deriving urban CO emissions and evaluating emission inventories if it can be demonstrated that there is limited or no contamination from open burning of biomass.

This is a very valid point which we previously only briefly mentioned in our Appendix. We have added a more elaborate explanation of how we deal with fires in Section 2.4 as well as the filter's effect in Appendix B.

Section 2.4:

*On the spatial scale relevant to plumes observed by TROPOMI, there can be contamination of the city signal by carbon monoxide produced by open fires (e.g. agricultural fires or wildfires). CO enhancements caused by open fires can result in overestimation of either the background or the downwind urban enhancement depending on their location. To avoid this, days with considerable CO contributions from open fires have been removed from our estimates. These days were selected based on the fire emission data from the Global Fire Assimilation System fire emission (GFAS) inventory (Kaiser et al., 2012) that is based on satellite measurements of fire radiative power. Days with cumulative fire emissions over 57 Mg per hour (equivalent to 0.5 Tg $yr^{-1}$) within $1.5°$ from the city center are removed. Additionally, days with strong burning events closer to the city (23 Mg $hr^{-1}$ within a $0.75°$ radius) are removed as well (Appendix B). Although the change in emission rate by this filtering is*

*limited for most cities, the filter can change estimated emission rates by up to 47%, as seen in Lusaka (Zambia).*

Appendix B:

*Regions with fewer estimates tend to be coastal. For example, we only have estimates for 160 days over Lagos and 113 for Dakar because of limited TROPOMI coverage over water.* **An additional reason for a small number of valid estimates lies in the occurrence of open fires; for example, 224 orbits (42%) are removed from our estimate over Lusaka (Zambia) due to fires within $1.5°$ of the city center and stronger fires within $0.75°$.**

**Review 2:**

Leguijt et al. presented their emission estimates of CO over several African cities based on TROPOMI observations and a computationally efficient cross-sectional flux (CSF) method. They presented methods for identifying the plume areas, calculating CO enhancements and CO fluxes, and for validating the CSF approach. They found discrepancies between space-based estimates and inventory-based estimates and leveraged emission ratios (ER) of CO to $CO_2$ from sector-specific emission inventories for further clues. An interesting "weekend effect" has also been revealed using CSF and TROPOMI XCO data.

General comments:

African cities or African land, in general, are somewhat mysterious and challenging to study from both the carbon budget and air pollution perspective. Hence, this study provides some interesting insights into the emissions of those rapidly growing cities, which could now be made possible using satellites monitoring the entire globe. Overall, this paper provided details and figures to support the descriptions of their plume detection and CSF methods. Aware of the potential limitations associated with the simplified CSF method, the authors conducted analyses to quantify emission uncertainties and reveal the flux threshold using the CSF method. The emission results are nicely presented to clearly show regional-specific differences between top-down (TD) and bottom-up (BU) estimates.

We thank the reviewer for the positive words.

Yet, based on the current presentation, it took me longer to comprehend the validation section (Sect. 2.6), e.g., the purposes of using WRF and what model quantities were the authors trying to optimize/evaluate without reading the previous studies in-depth. Although many ingredients of this paper were built on Varon et al. (2018), it would be more accessible to general readers if the authors could show how each obs/modeling component in Sect. 2 is related to one another since several different model products/variables were used for different purposes (e.g., TROPOMI, CSF, WRF, U10, Ueff, U from NECP, GEOS-FP, and ERA5...). For instance, having an overall flow chart may be helpful in showing what was derived from CSF, what was fed into/obtained from WRF simulations, and adjustments from Varon 2018 for urban emission estimates (e.g., a better characterization of wind fields). Detailed confusion about the validation procedure is included in the specific comments.

This is a point that came up in both reviews. We have included a flowchart as suggested which will explain our method more clearly and improved the methodology section as a whole.

*This section describes the different data products used in development of the Cross-Sectional Flux method and the simulations that were used to calibrate the model **and evaluate its performance using an Observing System Simulation***

*Experiment (OSSE). An OSSE is an experiment where a model or method is applied to synthetic data to evaluate the benefit of using this data and/or method. Which for this work means evaluating whether the CSF can be used to correctly estimate emissions from TROPOMI-like synthetic data. Figure 1 shows the roles of the different data products that are used and further described in Section 2.1 to 2.6. In addition, in Section 2.6 we show that the CSF method can be successfully applied to simulated data.*

[Figure]

*Figure 1. Schematic description of the use of the different data products within the OSSE and the subsequent emission quantification using TROPOMI data. The data products are discussed in the Section 2.1-2.6. First, the CSF is applied to simulated synthetic plumes in order to determine appropriate values for the various parameters used in our method. Second, an effective wind is calibrated by using the known emission rates of the simulated plumes following the procedure by Varon et al. (2018). The CSF, now calibrated on the synthetic plumes, is subsequently applied to satellite data to estimate emission rates of African cities.*

Three minor comments include 1) the lack of discussion of impacts from wildfire emissions and 2) secondary CO productions and CO sinks, and 3) possible discrepancy in the spatial extents of emissions from BU versus TD estimates (e.g., Fig. 6).

Wildfire and biofuel combustions play significant contributions to XCO signals and combustion efficiency over the African land. It is likely that influences from wildfire and chemical sources and sinks are minimized by the subtraction of the background XCO. However, the authors should provide some supplementary materials or investigations by, for example, examining wildfire inventories or satellite-based burned areas during the study periods. It helps verify whether TD CO emissions from TROPOMI for certain cities are affected by nonanthropogenic emissions, which may lead to systematic differences from the BU fossil fuel estimates (since pyrogenic emissions usually have relatively higher $CO/CO_2$ ERs than most FF emissions).

We have added an explanation of how we deal with fires in Section 2.4 (most notably by using fire CO emission data from the global fire assimilation system (GFAS) emission inventory) and the effect of the filtering on the number of days we can consider in Appendix B. We have expanded Section 2.3 and Section 2.5 to explain why chemical sources and sinks only have limited effect on the emissions estimated using our method.

Section 2.4:

*On the spatial scale relevant to plumes observed by TROPOMI, there can be contamination of the city signal by carbon monoxide produced by open fires (e.g. agricultural fires or wildfires). CO enhancements caused by open fires can result in overestimation of either the background or the downwind urban enhancement depending on their location. To avoid this, days with considerable CO contributions from open fires have been removed from our estimates. These days were selected based on the fire emission data from the Global Fire Assimilation System fire emission (GFAS) inventory (Kaiser et al., 2012) that is based on satellite measurements of fire radiative power. Days with cumulative fire emissions over 57 Mg per hour (equivalent to 0.5 Tg yr$^{-1}$) within 1.5° from the city center are removed. Additionally, days with strong burning events closer to the city (23 Mg hr$^{-1}$ within a 0.75° radius) are removed as well (Appendix B). Although the change in emission rate by this filtering is limited for most cities, the filter can change estimated emission rates by up to 47%, as seen in Lusaka (Zambia).*

Appendix B:

*Regions with fewer estimates tend to be coastal. For example, we only have estimates for 160 days over Lagos and 113 for Dakar because of limited TROPOMI coverage over water.* **An additional reason for a small number of valid estimates lies in the occurrence of open fires; for example, 224 orbits (42%) are removed from our estimate over Lusaka (Zambia) due to fires within 1.5° of the city center and stronger fires within 0.75°.**

Section 2.3:

*While EDGAR and DACCIWA only include the primary production of CO, the concentrations observed by TROPOMI include CO from secondary production as well. CO is produced by oxidation of volatile organic compounds (VOC) with methane as main contributor (Rozante et al., 2017). Mixing ratios of non-methane volatile organic compounds (NMVOC) observed in urban locations are typically of the order of $10^{-3}$–$10^{-2}$ [NMVOC]/[CO] (Von Schneidemesser et al., 2010). Dekker et al. (2019) showed that chemical production of CO by methane and NMVOC over cities only contribute 4% to the total CO signal, justifying the*

*simulation of CO as an inert tracer in our approach. Due to the 10 year atmospheric lifetime of methane, its contribution to CO production will result in a uniform concentration (Park et al., 2013), that is subtracted with the background. NMVOC have lifetimes of 0.6-10 days (Guo et al., 2007) that are much shorter than the lifetime of $CH_4$, but due to their low urban mixing ratios ($\sim$1%) their effect on the estimated emission rate is much smaller than the reported uncertainty of the CSF. This is consistent with the observation that the emission estimates of individual transects (that span a timescale of up to $\sim$10 hours) are stable and do not increase with increasing distance from the city (Section 2.4).*

It is unclear if TD emissions derived from different TROPOMI overpasses represent roughly the same spatial extent of the selected grid cells from the two inventories (which may affect BU estimates). To yield an apple-to-apple comparison between the TD and BU estimates, one needs to ensure the spatial extents represented by the two perspectives are similar OR the TD vs. BU differences (e.g., in the map of Figure 6) are not sensitive to how the authors selected the inventory grid cells.

This is a point which did not stand out sufficiently in our manuscript. We have therefore added a reference in our results to Section 2.2, where we explain how we made sure we make a fair comparison between TD and BU.

*As discussed in Section 2.2, we used different sizes for the city masks applied to the bottom-up inventories to ensure a fair comparison to the satellite-based emission estimates and found the choice of city mask did not impact our conclusions.*

Specific comments:

L93: "To test and calibrate our emission quantification approach we apply it to simulated data." – unclear. What do "it" and "simulated data" stand for?

We have rephrased this sentence to avoid confusion.

*To test and calibrate our emission quantification approach we apply **our CSF method** to simulated **TROPOMI** data **for three urban areas.***

L94: What did the authors mean by "simulate emissions"? Should it be "simulate column CO concentrations/mixing ratios"?

This is correct and we have changed the wording as suggested.

*We use the Weather Research and Forecasting (WRF) chemical transport model version 4.1 (Powers et al., 2017), to simulate **column CO mixing ratios** over Cairo (Egypt),*

L168: TROPOMI XCO AKs are accounted for in the WRF-based simulations. However, how TROPOMI XCO averaging kernel profiles were accounted for in the CSF method is not super clear. Typical XCO AKs deviate from 1 towards the surface.

We have added why the averaging kernel will have limited effect, and how we tested whether the CSF is able to give correct emissions estimates even when the AK is taken into account.

*A set of synthetic TROPOMI observations is created by sampling the simulation output over the TROPOMI footprints, applying its averaging kernel, selecting pixels based on quality value as discussed in Section 2.1, and adding Gaussian noise with a standard deviation equal to the reported uncertainty of the respective TROPOMI pixel.* **The TROPOMI quality value filtering ensures relatively clear sky observations with good surface sensitivity.**

*After determination of the effective wind on plumes with idealized pressure profiles, we test the performance of the CSF on more realistically sampled plumes which include the TROPOMI* **quality value filtering and averaging kernel sensitivities as described in Section 2.1 to see whether the CSF can correctly quantify emission rates from synthetic observations with quality filtering and non-uniform vertical sensitivity. To test the method's sensitivity, we perform an additional effective wind calibration on these data. The resulting linear fit (a = 1.4, b = −0.85, $R^2$ = 0.23) yields similar results and shows the filtering has limited impact on the calibration, while the lower $R^2$ value reflects the larger variation in estimated emission rates.**

L 169 – L171: "...see whether the CSF can reproduce the emissions used as model input." - So, emissions Q estimated using CSF and TROPOMI (Sect. 2.4) were fed into WRF to produce modeled XCO that can be evaluated against TROPOMI XCO? Or CSF is used to calculate emissions based on pseudo-XCO created by WRF + priors (e.g., EDGAR) like an OSSE experiment? What do "idealized" pressure weighted columns mean?

We have added a flowchart and additional text (as shown before) to explain the relation between the different datasets and that we are indeed performing an OSSE to calibrate our method. We have also added a sentence to explain the idealized pressure weighted columns.

*We also calculate "idealized" pressure weighted columns,* **which assume a uniform vertical sensitivity (flat averaging kernel),** *over the TROPOMI footprints without* **taking into account whether there is a valid** *TROPOMI* **observation** *as a first check to see whether the CSF can reproduce the emissions used as model input.*

L174: "Parameters like the width of the transects are tuned to get optimal quantification estimates on the simulated data" – I guess my confusion is still

related to the previous comment. I might miss something here, but how could the authors determine when the estimates are "optimal", especially when both CSF and WRF provide modeled values, not true observations? For example, the WRF 10m wind may not be accurate. Were there any observed wind observations that could be leveraged?

We have added a better explanation of how we made use of the WRF simulation, and why the WRF 10-m winds can be considered the truth for the purpose we are using them for.

> *We first test the validity of the CSF method using the idealized columns with 10-m winds output by the WRF simulation.* **The WRF winds are directly responsible for transport within the simulation, and can therefore be considered as the true wind fields behind the modeled concentrations.** *Parameters like the* **number** *of transects* **and distance of the background region** *are tuned to get optimal quantification estimates on the simulated data***, such that the fitted splines capture the observed curvature of the plumes and the background is not affected by the urban emissions.** *A list of the different parameters and their values can be found in Appendix A.*

> L175-L177: "We then use the simulations to calibrate the CSF method following the procedure by Varon et al. (2018). The wind speed in Eq. 2 is replaced by an effective wind speed..." – Without reading Varon2018, readers may be confused by the sudden introduction of effective wind speed (Ueff not mentioned in Sect. 2.4). Also, did the authors end up using the U(x, y) in Eq.2 or the alternative Ueff? I would suggest providing some context to this Ueff and to the calibration procedure in Varon2018 (e.g., what it was designed for).

The introduction of the effective wind was indeed rather brief. We have added text and rephrased to give a better explanation of the introduction of the effective wind speed and its purpose.

> **While the true wind field varies with altitude, the CSF method requires just a single (2D) wind field that is representative for the transport of the plume.** *We use the simulations to calibrate the CSF* **by introducing an effective wind speed that replaces the wind speed in Eq. 2** *following the procedure by (Varon et al., 2018).* **The effective wind speed is the wind that best captures the transport of the plume. It is a parametrization of the true** *wind speed to account for the effects of turbulence and variation in vertical wind speed and injection height. As the emission rates in the WRF simulations are known, the effective wind can be calculated* **explicitly** *for every orbit for each of the simulated cities.*

> L217-218: What does "concentrated emissions" mean? Please reword.

We have rephrased this term to avoid confusion.

> *EDGAR does not include any* **major** *emission* **sources** *around these cities*

L223-225: Does the regional inventory DACCIWA have a higher CO/CO$_2$ ER than EDGAR, since DACCIWA seems to agree better with TROPOMI-based estimates?

This is indeed the case for these cities, we have added this to our explanation of the difference between EDGAR and TROPOMI.

> This is further confirmed by **the higher CO/CO$_2$ values in DACCIWA and** the fact that the absolute CO$_2$ emission rates for these cities agree well between the two inventories.

Fig. 7: What are the emission sectors plotted in Fig 7? Only fossil fuel sectors or does it include other non-FF anthropogenic sectors (e.g., biofuel also with typical high CO/CO$_2$ ERs)? The EDGAR-based ERs appear to be always more variable than DACCIWA-based ratios. Would the choice of spatial extent affect those spatially mean ERs?

We have added text to better reflect the fact that we indeed compare to the sum of FF and non-FF anthropogenic emissions. The choice of spatial extent does have a limited effect on the CO inventory estimates, as explained in Section 2.2, but not to such an extent that it affects the patterns that we have observed. Similarly, the CO$_2$ inventory estimates vary based on the choice of city-boundaries, but the observed ER patterns (Northern Africa and South Africa having very low ERs compared to the rest of the cities in EDGAR) are unaffected.

> Each marker represents a single city. **The CO$_2$ values for both inventories include both fossil fuel and biofuel combustion emissions.** As power plants hardly emit any CO per kg of emitted CO$_2$ **due to their high combustion efficiency,** the contributions of this sector are removed from the CO$_2$ values.
>
> **As discussed in Section 2.2, we used different sizes for the city masks applied to the bottom-up inventories to ensure a fair comparison to the satellite-based emission estimates and found the choice of city mask did not impact our conclusions.**

L239 – 240: Why modeled winds like NCEP and GEOS-FP become important when discussing results for Nigeria, but not for other regions?

We have moved this comparison to Section 2.6 as it was indeed out of place in L239-240.

> In Lagos we estimate emissions of 0.36 (0.23-0.56) Tg yr$^{-1}$, that are consistent with EDGAR, but DACCIWA has emissions that are 5.2 times higher**, a difference which is much larger than the uncertainty in wind data discussed in Section 2.6.**

Fig 8: interesting to see the huge inventory-inventory discrepancy in emissions over Lagos. L279 – 280: "We evaluated the CSF method by applying it to a full-year of WRF simulations over three distinctly different African cities (Cairo, Lagos, and Bamako)." – Again, it would be more helpful to summarize what from

CSF has been evaluated? Is it to evaluate CSF's wind representations like U(x, y) or Ueff, its hyperparameters like # of transects, or its general capability in retrieval emissions (i.e., related to the simplified formula in Eq. 2 vs. full-physical models like WRF)?

We have rephrased this sentence to (together with the flowchart) better reflect what was done.

*We adapted and calibrated the computationally efficient Cross-Sectional Flux (CSF) method to quantify urban carbon monoxide emission rates from major cities in Africa using TROPOMI data. We **determined optimal values for the parameters of** the CSF by applying **the method** to a full-year of **simulated** WRF **plumes** over three distinctly different African cities (Cairo, Lagos, and Bamako**), such that the transects drawn best match the shape and curvature of the simulated plumes.** These simulations were also used to calibrate the CSF's effective wind speed relationship for TROPOMI data. **By applying the calibrated CSF to the simulated data with known emission rates, we** found that we can quantify urban CO emissions down to 0.1 Tg yr$^{-1}$ within 30% uncertainty.*